



# Collection/Aggregation in a Lagrangian cloud microphysical model: Insights from column model applications using LCM1D (v0.9)

Simon Unterstrasser[1], Fabian Hoffmann[2,3], and Marion Lerch[1]

[1]Deutsches Zentrum für Luft- und Raumfahrt (DLR) – Institut für Physik der Atmosphäre, Oberpfaffenhofen, 82234 Wessling, Germany.
[2]Cooperative Institute for Research in Environmental Sciences (CIRES), University of Colorado Boulder, Boulder, Colorado, USA
[3]NOAA Earth System Research Laboratory (ESRL), Chemical Sciences Division, Boulder, Colorado, USA

**Correspondence:** Simon Unterstrasser: simon.unterstrasser@dlr.de

**Abstract.** Lagrangian cloud models (LCMs) are considered the future of cloud microphysical modeling. However, LCMs are
computationally expensive due to the typically high number of simulation particles (SIPs) necessary to represent microphysical
processes such as collection/aggregation successfully. In this study, the representation of collection/aggregation is explored in
one-dimensional column simulations, allowing for the explicit consideration of sedimentation, complementing the authors'
previous study on zero-dimensional collection in a single grid box. Two variants of the Lagrangian probabilistic all-or-nothing
(AON) collection algorithm are tested that mainly differ in the assumed spatial distribution of the droplet ensemble: The first
variant assumes the droplet ensemble to be well-mixed in a predefined three-dimensional grid box (WM3D), while the second
variant considers explicitly the vertical coordinate of the SIPs, reducing the well-mixed assumption to a two-dimensional,
horizontal plane (WM2D). Since the number of calculations in AON depends quadratically on the number of SIPs, an approach
is tested that reduces the number of calculations to a linear dependence (so-called linear sampling). All variants are compared
to established Eulerian bin model solutions. Generally, all methods approach the same solutions, and agree well if the methods
are applied with sufficiently high accuracy (foremost the number of SIPs, timestep, vertical grid spacing). However, it is found
that the rate of convergence depends on the applied model variant. The dependence on the vertical grid spacing can be reduced
if AON WM2D is applied. The study also shows that the AON simulations with linear sampling, a common speed-up measure,
converges slower, as smaller timesteps are required to reach convergence compared to simulations with a quadratic dependence
on the number of SIPs. Most importantly, the study highlights that results generally require a smaller number of SIPs per grid
box for convergence than previous box simulations indicated. The reason is the ability of sedimenting SIPs to interact with
an effectively larger ensemble of particles when they are not restricted to a single grid box. Since sedimentation is considered
in most commonly applied three-dimensional models, the results indicate smaller computational requirements for successful
simulations than previously assumed, encouraging a wider use of LCMs in the future.





## 1 Introduction

Clouds are a fundamental part of the global hydrological cycle, responsible for the transport and formation of precipitation. While we expect a global increase in precipitation due to climate change, our knowledge on its spatial redistribution, including decreasing rainfall in some regions of the globe, is still uncertain (Boucher et al., 2013). The formation processes of precipitation are, however, reasonably understood and contain mechanisms that increase the size of hydrometeors. For liquid clouds, the coalescence of smaller cloud droplets is essential to form precipitating raindrops. In ice clouds, diffusional growth can produce precipitation-sized particles. The aggregation of ice crystals into larger clusters, snowflakes, also occurs frequently. And in mixed-phase clouds, ice crystals accrete supercooled liquid droplets forming graupel or hailstones.

The representation of these microphysical processes in climate models is impelled by the available computational resources, requiring necessary idealizations. Primarily, this is the case for computationally efficient Eulerian bulk models that predict only a small number of statistical moments for each hydrometeor class (e.g., Kessler, 1969; Khairoutdinov and Kogan, 2000; Seifert and Beheng, 2001), with commensurate effects on the representation of clouds and precipitation. Of course, more detailed cloud microphysics models have been also developed: Eulerian bin models represent cloud droplets on a mass grid that consists of hundreds of bins sampling the droplet size distribution (DSD) (e.g., Berry and Reinhardt, 1974; Tzivion et al., 1987; Bott, 1998; Simmel et al., 2002; Wang et al., 2007). But even these models exhibit limitations and idealizations. For instance, the coalescence of droplets is modeled as a Smoluchowski (1916) process, describing the mean evolution of an infinitely large, well-mixed droplet ensemble. The underlying Smoluchowski equation (also called the kinetic collection equation or even the stochastic collection equation, although the equation is deterministic), however, inherently neglects correlations and stochastic fluctuations known to be an integral part of the process chain that leads to precipitation (Gillespie, 1972; Bayewitz et al., 1974; Kostinski and Shaw, 2005; Wang et al., 2006; Alfonso et al., 2008).

In the last decade, Lagrangian cloud models (LCMs) emerged as a valued alternative to bin models for the detailed modeling of clouds (e.g., Andrejczuk et al., 2008; Sölch and Kärcher, 2010; Shima et al., 2009; Riechelmann et al., 2012; Arabas et al., 2015; Naumann and Seifert, 2015; Hoffmann et al., 2019). These models use Lagrangian particles, so-called simulation particles (SIPs) (Sölch and Kärcher, 2010) or superdroplets (Shima et al., 2009), each representing an ensemble of identical real droplets. Collection and aggregation in LCMs has recently been rigorously evaluated in box model simulations by Unterstrasser et al. (2017) (abbreviated as U2017 in the following), who compared three approaches documented in the literature: the remapping algorithm (RMA) by Andrejczuk et al. (2010), the average-impact algorithm (AIM) by Riechelmann et al. (2012), and the all-or-nothing algorithm (AON) developed by Shima et al. (2009) and Sölch and Kärcher (2010). RMA and AIM are deterministic algorithms and, in theory, approach the Smoluchowski solution of a reference bin model. The actual convergence of the algorithm, however, was found to depend significantly on properties of the SIP ensemble and the chosen kernel. The probabilistic AON indicated much better convergence properties, when it was averaged over sufficiently many instances. Furthermore, Dziekan and Pawlowska (2017) showed that AON approximates the stochastically complete Master equation including aforementioned correlations and stochastic fluctuations (Gillespie, 1972; Bayewitz et al., 1974). In fact, AON solutions





**Table 1.** List of abbreviations. **(Am Ende Text nochmal durchgehen um auch durchgehend die Abkuerzungen zu benutzen)**

| | |
|---|---|
| AON | All-or-nothing algorithm |
| BC | boundary condition |
| DSD | Droplet size distribution |
| GB | Grid box |
| LCM | Lagrangian cloud model |
| LWC | Liquid wter content |
| SIP | Simulation particle |
| U2017 | Unterstrasser et al. (2017) |

are identical to the Master equation solutions when the weighting factors (the number of real droplets represented by a SIP) are
set to unity.
However, many aspects of this relatively young modeling approach have not been tested thoroughly. One important message
of our previous box simulations in U2017 aws that the representation of collection exhibits considerably more freedom in
setting up a simulation than in bin models. Accordingly, in this study, we are going to extend the box simulations of U2017 by
analyzing collection in a vertical column, including sedimentation, as it has been done in previous studies for Eulerian bulk and
bin models (e.g., List et al., 1987; Tzivion (Tzitzvashvili) et al., 1989; Hu and Srivastava, 1995; Prat and Barros, 2007; Stevens
and Seifert, 2008; Seifert, 2008). All simulations will use the AON collection algorithm since it outperformed RMA and AIM
in the box simulations, and we do not expect that this general behavior is reversed here. The simulations will be compared
to established Eulerian bin references. Note that although the following analysis focuses on cloud droplets, the results can be
generalized for the LCM representation of ice crystal aggregation and the accretion of supercooled droplets. Therefore, we will
use the term collection to address coalescence, aggregation, or accretion as we will focus on the numerical treatment, which is
similar for all three process, and not on the physics. Moreover, we will use the term cloud droplets interchangeably with ice
crystals to increase clarity in writing.
The paper is structured as follows. First, Sec. 2 will give an overview on applied models, their foundations, and basic
setup. The results are presented in Sec. 3, divided into validation studies (Sec. 3.1), highly idealized applications in which the
column model emulates a box model (Sec. 3.2), process-level analysis of the applied algorithms (Sec. 3.3), and finally realistic
applications (Sec. 3.4). The paper is concluded in Sec. 4.
**2   Numerical model and setup**
Two column models which consider collection and sedimentation have been implemented, the first one represents a traditional
Eulerian bin scheme and the second model uses a particle-based approach. Before we describe both models in some detail,
we will (sometimes pedantically) write out basic relations, which will help disentangling the effects of particular parameter
variations later.





## 2.1 Basic relations and definitions

We use a column with $nz$ grid boxes (GBs). Each GB has the volume $\Delta V$ and a height of $\Delta z$. The total column height is thus

$$Lz = nz \times \Delta z. \tag{1}$$

We define that the GB $k$ with $1 \leq k \leq nz$ extends from $z_{k-1}$ to $z_k := k \times \Delta z$, hence the GB with $k = 1$ is the lowest GB.

The horizontal area of the column is given by

$$\Delta A = \Delta V / \Delta z. \tag{2}$$

The droplets are assumed to be spherical with a density of $\rho_w = 1000\,\mathrm{kg/m^3}$ and the mass-size relation is simply given by

$$m = \frac{4}{3}\pi r^3 \rho_w. \tag{3}$$

Following Gillespie (1972) and Shima et al. (2009), the probability $P_{ij}^{WM3D}$ that one droplet with mass $m_i$ coalesces with one droplet with mass $m_j$ inside a small volume $\delta V$ within a short time interval $\delta t$ is given by

$$p_{ij}^{WM3D} = K_{ij}\,\delta t\,\delta V^{-1}, \tag{4}$$

where $K_{ij} = K(m_i, m_j)$ or equivalently $K(r_i, r_j)$. We suppose that $\delta t$ is sufficiently small in order to assure $p_{ij}^{WM3D} \leq 1$.

The hydrodynamic collection kernel is given by

$$K^{WM3D}(r_i, r_j) = E_c(r_i, r_j)\pi(r_i + r_j)^2\,|w_{sed,i} - w_{sed,j}|, \tag{5}$$

where $w_{sed}$ is the radius-dependent droplet fall speed and $E_c = E \times E_{\mathrm{coal}}$ is the collection efficiency, which is the product of the collision efficiency $E$ and the coalescence efficiency $E_{\mathrm{coal}}$. In this study, we use the $w_{sed}$-parametrisation of Beard (1976), the tabulated $E$-values of Hall (1980), and the coalescence efficiency $E_{\mathrm{coal}}$ is assumed to be 1. The latter assumption is an oversimplification for large droplets with radii $\gtrsim 500\,\mu\mathrm{m}$ for which $E_{\mathrm{coal}}$ is significantly smaller than 1 (Beard and Ochs III, 1984; Ochs III and Beard, 1984), but does not limit the generality of our findings.

The average number of collisions from $\nu_i$ droplets of mass $m_i$ and $\nu_j$ droplets of mass $m_j$ (which are assumed to be well-mixed in the volume $\delta V$) within time $\delta t$ is

$$\nu_{coll} = K_{ij}^{WM3D}\,\nu_i\,\nu_j\,\delta t\,\delta V^{-1}, \tag{6}$$

or equivalently

$$\nu_{coll} = E_c(r_i, r_j)\pi(r_i + r_j)^2\,|w_{sed,i} - w_{sed,j}|\nu_i\,\nu_j\,\delta V^{-1}\delta t. \tag{7}$$

By dividing the above equation by $\delta V$, we obtain the common relationship in terms of concentrations, given by $n = \nu/\delta V$,

$$n_{coll} = E_c(r_i, r_j)\pi(r_i + r_j)^2\,|w_{sed,i} - w_{sed,j}|n_i\,n_j\,\delta t. \tag{8}$$





Sedimentation and collection are the only processes considered in this study, and any effects of diffusional growth are
neglected. An exponential DSD is used to prescribe the cloud droplets in the beginning

$$f_{\mathrm{m}}(m) = \frac{DNC}{\bar{m}} \exp\left(-\frac{m}{\bar{m}}\right). \tag{9}$$

As in U2017, Berry (1967) or Wang et al. (2007), we choose by default a mean mass $\bar{m} = LWC/DNC$ that corresponds
to a mean droplet radius of $r_0 = 9.3\,\mu\mathrm{m}$ and a droplet number concentration $DNC = 2.97 \times 10^8\,\mathrm{m}^{-3}$ (resulting in a droplet
mass concentration of $LWC = 10^{-3}\,\mathrm{kg\,m^{-3}}$). The function $f_{\mathrm{m}}(m)$ is the number density function with respect to mass. The
moments are defined as

$$\lambda_l(t) = \int m^l f_{\mathrm{m}}(m, t)\mathrm{d}m, \tag{10}$$

with order $l$, which gives $DNC = \lambda_0, LWC = \lambda_1$ and $Z = \lambda_2$. We will refer to the latter quantity as radar reflectivity since
the radar reflectivity is proportional to $\lambda_2$. For an exponential DSD, the moments can be expressed analytically as

$$\lambda_{l,\mathrm{anal}} = (l-1)!\,DNC\,\bar{m}^l, \tag{11}$$

where $l!$ is the factorial of $l$.
Using the terminology of Berry (1967), we introduce the mass density function with respect to the logarithm of droplet
radius $\ln r$

$$g_{\ln r}(r) = 3m^2 f_{\mathrm{m}}(m), \tag{12}$$

taking into account the transformation property of distributions ($f_{\mathrm{y}}(y)\mathrm{d}y = f_{\mathrm{x}}(x(y))\mathrm{d}x$).
The DSD is usually discretised using exponentially increasing bin sizes. In analogy to U2017, the bin boundaries are defined
by the masses

$$m_{bb,p+1} = m_{bb,p}\,10^{1/\kappa}. \tag{13}$$

Note that many other studies use a factor of $2^{1/s}$ for discretisation. The parameters $s$ and $\kappa$ are related via $s = \kappa \log_{10}(2) \approx$
$0.3\,\kappa$.
In an LCM, real droplets are represented by simulation particles (SIPs, also called super droplets). Each SIP has a discrete
position (vertical coordinate $z_p$ in our column model applications) and represents $\nu_p$ identical real droplets with an individual
droplet mass $\mu_p$. The total droplet mass in a SIP is then $\nu_p\mu_p$. In conjunction with SIPs, we define that the terms low and high
relate to the SIP vertical position and the terms small and large to the droplet mass $\mu_p$. The number of SIPs in a GB is defined
as $N_{SIP,GB}$ and the total SIP number is given by $N_{SIP,tot} = \sum_{k=1}^{nz} N_{SIP,GB}(k)$.
The moments $\lambda_l$ of order $l$ in a GB are computed via a simple summation

$$\lambda_{l,\mathrm{SIP}} = \left(\sum_{p=1}^{N_{\mathrm{SIP,GB}}} \nu_p\,\mu_p^l\right) \Big/ \Delta V, \tag{14}$$



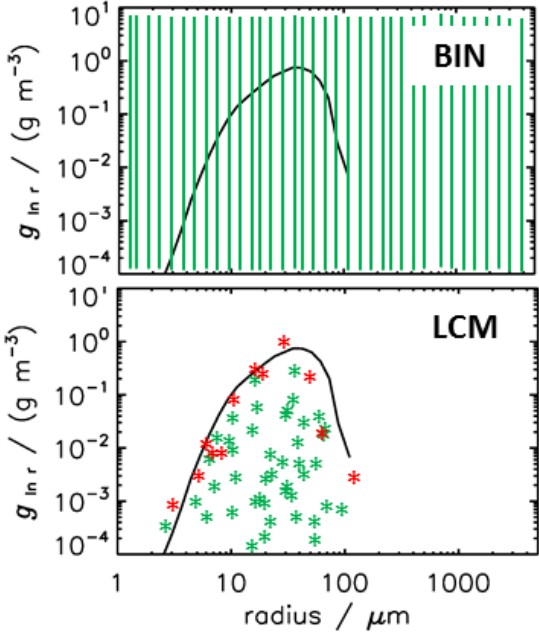

**Figure 1.** schematic plot of how a droplet dize distribution is discretized in a bin model and represented by a SIP (SImulation particle) ensemble in a Lagrangian cloud model (LCM). The red and green stars shows two different realisations of a SIP ensemble.

Here and in the following, index $p$ refers to any single bin or SIP. If we want to stress that the combination of two SIPs or bins
matters, we use indices $i$ and $j$. Index $k$ is used for altitude and $l$ for the order of the moments by convention.

3       How to represent an ensemble of droplets in an Eulerian or Lagrangian cloud model? Their size distribution can be uniquely

described in a bin model by simply accounting for each real droplet in its respective bin, where its boundaries are given by the
bin model (see illustration in Fig. 1 top). In the Lagrangian approach, however, the weighting factor $\nu_i$ and the droplet mass $\mu_i$
can be chosen independently. Accordingly, there is no unique SIP representation of an ensemble of real droplets; two possible
SIP ensemble realisations are illustrated in Fig. 1 bottom.

8       Various techniques to generate a SIP ensemble in an LCM for a given (analytically prescribed) DSD exist (see section 2.1 in

U2017). In this study, we use a SIP initialisation technique (termed "singleSIP-init" in U2017), for which Lagrangian collection
algorithms, and in particular AON, achieved the best results in box model tests. In the singleSIP-init, the DSD, more specifically
$f_m$, is discretized in exponentially increasing mass intervals and a single SIP is generated for each bin (see section 2.1.1 in
U2017 for details). The SIP weight is given by
$\nu_{\mathrm{p}} = f_{\mathrm{m}}(\mu_{\mathrm{p}}) \, \Delta m_{\mathrm{bb},p} \Delta V,$       (15)
where $\mu_{\mathrm{p}}$ is chosen randomly from the interval $[m_{\mathrm{bb},p}, m_{\mathrm{bb},p+1}]$. The generation of SIPs with $\nu_{\mathrm{p}}$ below some threshold is
discarded. Due to the probabilistic component, different realisations of SIP ensembles can be created for the same prescribed
DSD, yet the init technique guarantees that the moments $\lambda_{l,\mathrm{SIP}}$ are close to $\lambda_{l,\mathrm{anal}}$. The number of generated SIPs depends on





the width of the mass bins and hence on $\kappa$, as well as the other parameters of the prescribed DSD. A change of the "system
size" $\Delta V$ does not change the number of SIPs, but simply leads to a rescaling of the SIP weights $\nu_i$. For exponential DSD
given above, around

$$N_{SIP,GB} = 5 \times \kappa \tag{16}$$

SIPs are initialised (the scaling factor depends on the width of DSD and the choice of the lower cut-off threshold). Finally note
that if the DSD is prescribed in a specific GB, the position $z_p$ of each SIP is randomly chosen from $[z_k, z_{k+1}]$. Furthermore, $\delta t$
and $\delta V$ of the conceptual model take the values $\Delta t$ and $\Delta V$ in the numerical models.
## 2.2 Eulerian column model
Eulerian column models have been widely employed in cloud physics and the present bin implementation is conceptually
similar to previous ones (e.g. Prat and Barros, 2007; Stevens and Seifert, 2008; Hu and Srivastava, 1995). We use exponentially
increasing bin sizes as defined in Eq. 13. The smallest mass $m_{bb,0}$ is chosen suitably small (corresponding roughly to a droplet
radius of $1\,\mu\mathrm{m}$), and the grid resolution parameter $s$ sufficiently large (4 by default), i.e. the mass doubles every four bins.
The variable $g_{\ln m} = \frac{1}{3} g_{\ln r}$ will be discretized in mass space and used as a prognostic variable. The droplet mass concentra-
tion in each bin $p$ and height $k$ is given by $g_{p,k} \times \mathrm{d}\ln m$ and approximates $\int_{m_{\mathrm{bb},p}}^{m_{\mathrm{bb},p+1}} g_{\ln m}(m, z_k)\mathrm{d}\ln m$. For each GB $k$, Bott's
exponential flux method (Bott, 1998, 2000) is used to solve the Smoluchowski. Bott's method is a one-moment scheme and
$g_{\ln m}$ is the only prognostic variable. In a second step, the mass concentrations are advected according to the classical advection
equation

$$\frac{d\, g_{\ln m}}{dt} = w_{sed} \frac{d\, g_{\ln m}}{dz}. \tag{17}$$

For its numerical solution, two different positive definite advection algorithms have been used. The first option is the classical
first-order upwind scheme (known for its inherent numerical diffusivity). For $w_{sed} \geq 0$, it is simply given by

$$g_{p,k}(t + \Delta t) = g_{p,k}(t) + \frac{\Delta t}{\Delta z} w_{sed}(\bar{m}_{\mathrm{bb},p})(g_{p,k+1}(t) - g_{p,k}(t)). \tag{18}$$

The above equation is solved independently for each bin $p$, where $w_{sed}$ is evaluated at the arithmetic bin center $\bar{m}_{\mathrm{bb},p} =$
$0.5\,(m_{\mathrm{bb},p+1} + m_{\mathrm{bb},p})$ [1] . A second (better) option is the popular MPDATA algorithm, which is an iterative solver based on
the upwind scheme, yet drastically reduces its diffusivity (Smolarkiewicz, 1984, 2006). By default, MPDATA is employed.
Irrespective of the chosen advection solver, the prediction of the "new" $g_{p,k}$ depends on $g_{p,k}$ and $g_{p,k+1}$ (i.e. the GB above
the one of interest). For the prediction of $g_{p,nz}$ at the model top, it is necessary to prescribe some value $g_{p,nz+1}$ which defines
the upper boundary condition (this is detailed in section 2.4).
If the prescribed $\Delta t$ is too large and the Courant-Friedrichs-Levy (CFL) criterion $\frac{\Delta t}{\Delta z} w_{sed}(\bar{m}_{\mathrm{bb},p}) \leq r_{CFL} < 1$ is violated,
subcyling is introduced. As $w_{sed}(\bar{m}_{\mathrm{bb},p})$ does not change over the course of a simulation, the (bin-dependent) number of
subcycles $n_{subc,p}$ is determined in the beginning, such that $r_{CFL} = 0.5$ holds for the reduced timestep $\frac{\Delta t}{n_{subc,p}}$.

---

[1] Evaluating $w_{sed}$ at the geometric bin centers did not change the results.



After one call of the Bott algorithm, $n_{subc,p}$ calls of the selected advection algorithm with reduced time step $\frac{\Delta t}{n_{subc,p}}$ follow
for each bin $p$.
The moments are computed by

$$\lambda_{l,\text{BIN}} = \sum_{p=1}^{N_{\text{BIN}}} g_{p,k} \, (\tilde{m}_{\text{bb},p})^{l-1} \frac{\ln 2}{3\,s} \tag{19}$$

as given in Eq. 48 of Wang et al. (2007), where $\tilde{m}_{\text{bb},p} = m_{\text{bb},p} \times 2^{1/(2\,s)}$ is the geometric bin center.

## 2.3  Lagrangian column model

In a Lagrangian model, the inclusion of sedimentation (obeying the transport equation $dz/dt = -w_{sed}$) is straightforward. For
each SIP the particle position is updated via

$$z_p(t + \Delta t) = z_p(t) - w_{sed}(\mu_p(t)) \, \Delta t. \tag{20}$$

Unlike to Eulerian methods, sedimentation in a Lagrangian approach is independent of the chosen mesh and the time step is
not restricted by numerical reasons. If $z_p$ becomes negative at some point in time, the SIP crossed the lower boundary and is
removed.
For the collection process, it assumed that each SIP belongs to a certain GB $k$ obeying $z_{k-1} \leq z_p < z_k$ and that the real
droplets of each SIP are well-mixed in the GB volume (WM3D). The collection process is treated with the probabilistic AON
algorithm. In the regular version (see section 2.3.1), AON is called for each GB and accounts for all possible collisions among
any two SIPs of the same GB. By construction, the information on the vertical position is irrelevant inside the regular AON,
and is only used in the SIP-to-GB assignment.
In the version with explicit overtakes (WM2D, see section 2.3.2), for any two SIPs (of the whole column) it is checked if
the higher SIP (i.e. with larger $z_p$) overtakes the lower SIP within the current time step. This may have several advantages:
First, only 2D well-mixedness in a horizontal plane is assumed and possible size sorting effects within a GB are accounted
for. Moreover, in Lagrangian methods the time step is not restricted by the CFL criterion and the largest SIPs may travel
through more than one GB. In the classical approach, such a SIP can only collect SIPs from the GB where it was present in the
beginning of the time step. In the second approach, collections can also occur across GB boundaries (see section 2.3.2).
In the remainder of this paper, the classical approach is referred to as "3D Well-Mixed" (WM3D) AON and the new approach
as AON-WM2D. Figure 2 sketches how the SIP properties (location, weighting factor, sedimentation speed) are interpreted in
either approach. For simplicity, a single GB with one SIP pair is displayed.
AON is probabilistic and an individual realisation does usually not reproduce the mean state as predicted by deterministic
methods like Eulerian approaches. The extent of deviations from the mean state is exemplified in Fig. 15 of U2017 for a
box model application of AON. Hence, the discussed AON results in the present study are usually ensemble averages over
$nr_{inst} = 20$ realisations.
Pseudo-code of both algorithm implementations is given. For the sake of readability, the pseudo-code examples show easy-
to-understand implementations. The actual codes of the algorithms are, however, optimised in terms of computational effi-





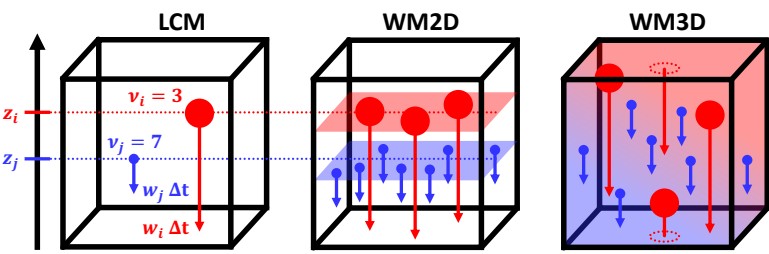

**Figure 2.** Grid box with a SIP pair in the LCM world (left) and its respective interpretation in the 2D Wellmixed (WM2D, center) and 3D Wellmixed (WM3D, right) approach of the AON collection algorithm.

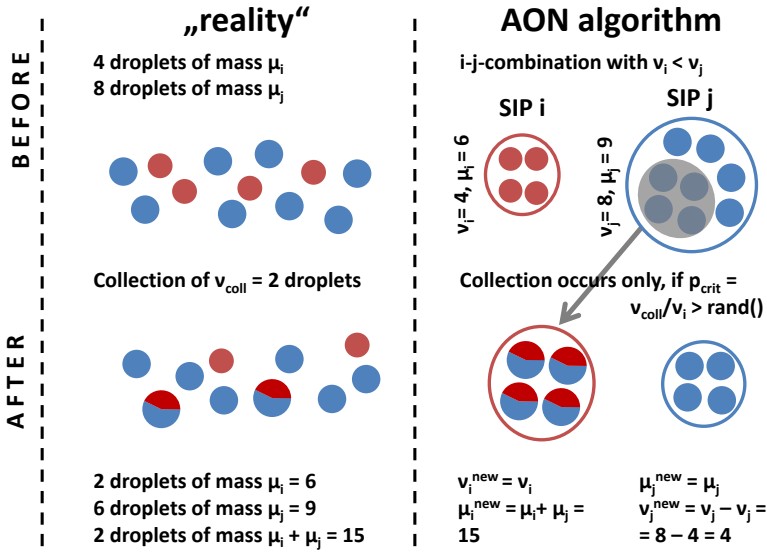

**Figure 3.** Treatment of a collection between two SIPs in the All-Or-Nothing Algorithm (AON) algorithm, adopted from Fig. 2 of Unterstrasser et al. (2017).

ciency. The style conventions for the pseudo-code examples are as follows: commands of the algorithms are written in upright
font with keywords in boldface. Comments appear in italic font (explanations are enclosed by { } and headings of code blocks
are in boldface).
**2.3.1 Regular AON collection algorithm (WM3D)**
Here we basically repeat the AON description of U2017 (their section 2.5).





---

**Algorithm 1** Pseudo-code of the WM3D all-or-nothing algorithm (AON); style conventions are explained right before Section 2.3.1 starts; rand() generates uniformly distributed random numbers $\in [0,1]$. This AON version is called independently for each grid box.

---

 1: *INIT BLOCK*

 2: Given: Ensemble of SIPs of a specific grid box;   Specify: $\Delta t$

 3: *TIME ITERATION*

 4: **while** t<Tsim **do**

 5:     *{Check each $i-j$-combination for a possible collection event}*

 6:     **for all** $i < j \leq N_{\mathrm{SIP}}$ **do**

 7:         Compute $\nu_{coll}$ according to Eq. 7

 8:         $\nu_{\mathrm{new}} = \min(\nu_i, \nu_j)$

 9:         $p_{\mathrm{crit}} = \nu_{coll}/\nu_{\mathrm{new}}$

10:         *{Update SIP properties on the fly}*

11:         **if** $p_{\mathrm{crit}} > 1$ **then**

12:             ***MULTIPLE COLLECTION***

13:             *{can occur when $\nu_i$ and $\nu_j$ differ strongly and be regarded as special case; see text for further explanation}*

14:             assume $\nu_i < \nu_j$, otherwise swap $i$ and $j$ in the following lines

15:             *{$p_{\mathrm{crit}} > 1$ is equivalent to $\nu_{coll} > \nu_i$}*

16:             *{transfer $\nu_{coll}$ droplets with $\mu_j$ from SIP $j$ to SIP $i$, allow multiple collections in SIP $i$, i.e. one droplet of SIP $i$ collects more than one droplet of SIP $j$.}*

17:             SIP $i$ collects $\nu_{coll}$ droplets from SIP $j$ and distributes them on $\nu_i$ droplets: $\mu_i = (\nu_i\,\mu_i + \nu_{coll}\,\mu_j)/\nu_i$

18:             SIP $j$ loses $\nu_{coll}$ droplets to SIP $i$: $\nu_j = \nu_j - \nu_{coll}$

19:         **else if** $p_{\mathrm{crit}} >$ rand() **then**

20:             ***RANDOM SINGLE COLLECTION***

21:             assume $\nu_i < \nu_j$, otherwise swap $i$ and $j$ in the following lines

22:             *{transfer $\nu_i$ droplets with $\mu_j$ from SIP $j$ to SIP $i$}*

23:             SIP $i$ collects $\nu_i$ droplets from SIP $j$: $\mu_i = \mu_i + \mu_j$

24:             SIP $j$ loses $\nu_i$ droplets to SIP $i$: $\nu_j = \nu_j - \nu_i$

25:         **end if**

26:     **end for**

27:     $t = t + \Delta t$

28: **end while**

---



*"Figure 3 illustrates how a collection between two SIPs is treated. SIP $i$ is assumed to represent fewer droplets than SIP $j$,*
*i.e. $\nu_i < \nu_j$. Each real droplet in SIP $i$ collects one real droplet from SIP $j$. Hence, SIP $i$ contains $\nu_i = 4$ droplets, now with*
*mass $\mu_i + \mu_j = 15$. SIP $j$ now contains $\nu_j - \nu_i = 8 - 4 = 4$ droplets with mass $\mu_j = 9$. Following Eq. (7), only $\nu_{coll} = 2$ pairs*
*of droplets would, however, merge in reality. The idea behind this probabilistic AON is that such a collection event is realised*
*only under certain circumstances in the model, namely such that the expectation values of collection events in the model and*
*in the real world are the same. This is achieved if a collection event occurs with probability*

$$p_{\text{crit}} = \nu_{coll}/\nu_i \tag{21}$$

*in the model. Then, the average number of collections in the model,*

$$\bar{\nu}_{coll} = p_{\text{crit}}\nu_i = (\nu_{coll}/\nu_i)\nu_i, \tag{22}$$

*is equal to $\nu_{coll}$ as in the real world. A collection event between two SIPs occurs if $p_{\text{crit}} >$ rand(). The function rand() provides*
*uniformly distributed random numbers $\in [0, 1]$. Noticeably, no operation on a specific SIP pair is performed if $p_{\text{crit}} <$ rand().*
*The treatment of the special case $\nu_{coll}/\nu_i > 1$ needs some clarification. This case is regularly encountered when SIPs with*
*large droplets and small $\nu_i$ collect small droplets from a SIP with large $\nu_j$. The large difference in droplet masses $\mu$ led to*
*large kernel values and high $\nu_{coll}$ with $\nu_i < \nu_{coll} < \nu_j$. [. . . ] If $p_{\text{crit}} > 1$, we allow multiple collections, as each droplet in*
*SIP $i$ is allowed to collect more than one droplet from SIP $j$. In total, SIP $i$ collects $\nu_{coll}$ droplets from SIP $j$ and distributes*
*them on $\nu_i$ droplets. A total mass of $\nu_{coll}\mu_j$ is transferred from SIP $j$ to SIP $i$ and the droplet mass in SIPs $i$ becomes $\mu_i^{\text{new}} =$*
*$(\nu_i\mu_i + \nu_{coll}\mu_j)/\nu_i$. The number of droplets in SIP $j$ is reduced by $\nu_{coll}$ and $\nu_j^{\text{new}} = \nu_j - \nu_{coll}$. Keeping with the example in*
*Fig. 3 and assuming $\nu_{coll} = 5$, each of the $\nu_i = 4$ droplets would collect $\nu_{coll}/\nu_i = 1.25$ droplets. The properties of SIP $i$ and*
*SIP $j$ are then $\nu_i = 4$, $\mu_i = 17.25$, $\nu_j = 3$ and $\mu_j = 9$. [. . . ] So far, we explained how a single $i - j$ combination is treated*
*in AON. In every time step, the full algorithm simply checks each $i - j$ combination for a possible collection event. To avoid*
*double counting, only combinations with $i < j$. Pseudo-code of the algorithm is given in Algorithm (1). The SIP properties are*
*updated on the fly. If a certain SIP is involved in a collection event in the model and changes its properties, all subsequent*
*combinations with this SIP take into account the updated SIP properties. [. . . ] For the generation of the random numbers, the*
*well-proven (L'Ecuyer and Simard, 2007) Mersenne Twister algorithm by Matsumoto and Nishimura (1998) is used."*
The AON treatment of self-collections and of SIPs with equal weighting factors are described in U2017. In the simulations
presented here these aspects are not relevant and thus omitted.
The current implementation differs in several aspects from the version in Shima et al. (2009). First, they use a linear sampling
approach (which will be described in subsection 2.3.3). Second, the weighting factors are considered to be integer numbers,
whereas we use real numbers $\nu$. Integer values are appropriate in discrete test cases of small sample volumes such as the
validation test case in section 3 of Dziekan and Pawlowska (2017). For comparing AON with bin model references, usually
continuous DSDs are prescribed. Then a SIP ensemble with real-values weighting factors is more appropriate. Third, multiple
collections (MC) are differently treated. For $p_{crit} = (\nu_{coll}/\nu_i) > 1$, either $\lfloor p_{crit} \rfloor \nu_i$ or $\lceil p_{crit} \rceil \nu_i$ droplets of SIP $j$ merge with
$\nu_i$ droplets of SIP $i$ depending on the probability $p_{crit} - \lfloor p_{crit} \rfloor$. This maintains the integer property of the SIP weights. As the





latter feature is not required in our approach, we deterministically merge $p_{crit}\nu_i = \nu_{coll}$ droplets from SIP $j$ with $\nu_i$ droplets
of SIP $i$. This is computationally more efficient than the integer-preserving implementation. Test simulations showed that both
MC treatments produce similar results.

### 2.3.2   AON algorithm with explicit use of vertical coordinate (WM2D)

We now introduce the AON version based on an idea by Sölch and Kärcher (2010) where the vertical position $z_p$ of the SIPs
is explicitly considered. The approach and its implications will be detailed next. Pseudo-code of this AON variant is given in
Algorithm 2.
Unlike to the classical case where 3D well-mixedness has to be assumed, droplets of a SIP are now assumed to be well
mixed on the x-y-plane at $z = z_p$ within the GB (horizontally well-mixed instead of the traditional isotropic assumption) and
represent a "concentration" of $n_{2D} = \nu/\delta A$ (units $L^{-2}$, where $L$ is a length scale). We introduce an adapted kernel definition
where the relative velocity term $|w_{sed,i} - w_{sed,j}|$ is dropped from Eq. 5:
$$K_{ij}^{WM2D} := E_c(r_i, r_j)\pi(r_i + r_j)^2. \tag{23}$$
The AON algorithm is split into two steps:
1. Based on the evaluation of the vertical positions $z_i$ and $z_j$ at times $t$ and $t + \Delta t$, it is checked if SIP $i$ overtakes SIP $j$
within a time step $\Delta t$. Given $z_i(t) \geq z_j(t)$ (otherwise swap $i$ and $j$) an overtake takes place in the time interval $\Delta t$ if
$z_i(t + \Delta t) < z_j(t + \Delta t)$.
2. In case of such an overtake: Compute the average number of droplet collections by
$$\nu_{coll} = K_{ij}^{WM2D}\nu_i\,\nu_j\,\Delta A^{-1}. \tag{24}$$
Analogous to the classical implementation, a collection in the model is performed with a probability $\nu_{coll}/\nu_i$ and SIP $i$
may collect $\nu_i$ from SIP $j$ (in this step $i$ and $j$ are chosen, such that $\nu_i < \nu_j$).
Similarly to the WM3D version, it happens that $\nu_{coll}$ is larger than $\nu_i$ and multiple collections should be considered in the
algorithm.
Specifically to WM2D, it is also possible that a SIP interacts with other SIPs located not only in one but several GBs.
Accordingly, it is not only necessary to check overtakes of other SIPs in the original GB (more specifically, SIPs that lie in the
same GB at time $t$), but also the SIPs that are located underneath, depending on the prescribed time step. In a Lagrangian model,
the time step choice is not numerically restricted by the CFL criterion and in particular the largest collecting drops may fall
through several GBs during the time period $\Delta t$. Hence, their collections are underrated unless potential overtakes are checked
among all $N_{SIP,tot}$ SIPs of the entire column. In a naive implementation this would dramatically increase the computational
costs. In the regular WM3D implementation, $nz$ calls of AON with $O(N_{SIP,GB}^2)$ (for simplicity lets assume $N_{SIP,GB}$ is the
same in all GBs) give a total cost of $nz \times O(N_{SIP,GB}^2)$. Contrarily, AON-WM2D is called once for all SIPs of the column.
Hence the cost is $1 \times O(N_{SIP,tot}^2) = nz^2 \times O(N_{SIP,GB}^2)$ and a factor $nz$ higher than the regular implementation. However,





**Algorithm 2** Pseudo-code of the WM2D all-or-nothing algorithm (AON); style conventions are explained right before Section 2.3.1 starts; rand() generates uniformly distributed random numbers $\in [0, 1]$. This AON version is called once for the total column.

1: ***INIT BLOCK***

2: Given: Ensemble of SIPs of the total column, in particular also their positions     Specify: $\Delta t$

3: ***TIME ITERATION***

4: **while** t<Tsim **do**

5:      *{Sort SIPs by position, the highest SIP will be the first SIP.}*

6:      Sort SIPs by position, such that $z_i(t) \geq z_j(t)$ for $i < j$

7:      *{Check for overtakes}*

8:      **for** $i = 1, N_{SIP,tot} - 1$ **do**

9:          **for** $j = i + 1, N_{SIP,tot}$ **do**

10:             **if** $z_i(t + \Delta t) \geq z_j(t)$ **then**

11:                 exit j-loop and proceed with next SIP $i$ *{if end position of SIP i is above departure point of SIPs j, then no overtakes are possible for any remaining SIP j.}*

12:             **end if**

13:             **if** $z_i(t + \Delta t) \geq z_j(t + \Delta t)$ **then**

14:                 proceed with next SIP $j$ *{no overtake occured as SIP i is still above SIP j at $t + \Delta t$}*

15:             **end if**

16:             *{the above conditions guarantee that the following code is executed iff SIP $i$ overtakes SIP $j$}*

17:             Compute $\nu_{coll}$ according to Eq. 24 *{ instead of Eq. 7 as in the WM3D version}*

18:             *{all the following operations are identical to the WM3D version and accompanying explanations are removed}*

19:             $\nu_{new} = \min(\nu_i, \nu_j)$

20:             $p_{crit} = \nu_{coll}/\nu_{new}$

21:             **if** $p_{crit} > 1$ **then**

22:                 assume $\nu_i < \nu_j$, otherwise swap $i$ and $j$ in the following lines

23:                 $\mu_i = (\nu_i \mu_i + \nu_{coll} \mu_j)/\nu_i$

24:                 $\nu_j = \nu_j - \nu_{coll}$

25:             **else if** $p_{crit} >$rand() **then**

26:                 assume $\nu_i < \nu_j$, otherwise swap $i$ and $j$ in the following lines

27:                 $\mu_i = \mu_i + \mu_j$

28:                 $\nu_j = \nu_j - \nu_i$

29:             **end if**

30:          **end for**

31:      **end for**

32:      $t = t + \Delta t$

33: **end while**





the WM2D implementation can be sped up by first sorting all SIPs by their position (if sorting is done independently in each
GB, the complexity is $nz \times O(N_{SIP,GB} \log(N_{SIP,GB}))$), and second by taking into account that the final position $z_i(t + \Delta t)$
of the potentially overtaking SIP $i$ must be below the initial position $z_j(t)$ of SIP $j$. Finding possible candidates for SIP $i$ within
the sorted SIP list can be stopped once a SIP $j$ with $z_j(t) < z_i(t + \Delta t)$ is encountered (see condition in line 10 of Algorithm 2).
For the smallest SIPs, which often travel only a small distance inside a GB, the list of SIPs that may be overtaken is com-
mensurately small and overtakes have to be checked for a fraction of SIPs of the GB only (that means the actual computational
work is smaller than in the regular version). On the other hand, imagine the largest SIPs travel through three GBs, then over-
takes have to be tested for roughly three times more SIPs than in the regular version. Moreover, testing for overtakes (step 1)
is computationally less demanding than calculating the potential collections (step 2). In WM3D we have always the workload
of step 2 for all tested combinations, whereas in WM2D only the cheaper step 1 is executed in case of no overtake.
Besides the weaker assumption of 2D well-mixedness, the present approach is actually more intuitive (even though it may
first be regarded counter-intuitive by those who are familiar with traditional Eulerian grid-based approaches). Moreover, this
approach complies better with the Lagrangian paradigm of a grid-free description (the present approach is independent of $nz$
and $\Delta z$, yet some horizontal "mixing area" $\Delta A$ has to defined, over which the droplets of a SIP are assumed to be dispersed).
For more sophisticated kernels, including, e.g., turbulence enhancement, the present approach may not be adopted easily
as the driving mechanism for collisions to occur in the current model is differential sedimentation (see also discussions on
cylindrical vs. spherical formulations of kernels in (Saffman and Turner, 1956) and Wang et al. (1998, 2005)).
Finally, we shortly summarize the differences between the WM2D and WM3D approach. The standard kernel $K^{WM3D}$ as
given by Eq. 5 has units $L^3/T$ (where $L$ and $T$ are a length and time scale, resp.). Multiplying it by concentrations $n_i$ and
$n_j$ (units $L^{-3}$) one obtains the rate of a concentration increase of merged droplets ($L^{-3}/T$) which is finally multiplied by $\delta t$
(unit $T$) to obtain $n_{coll}$ (see Eq. 8). Since SIPs represent droplet concentrations of $n_i = \nu_i/\delta V$ and $n_j = \nu_j/\delta V$, Eq. 7 follows.
In the WM2D approach, the kernel $K^{WM2D}$ as given by Eq. 23 has units $L^2$. Multiplying it by "2D" concentrations $n_{2D,i}$
and $n_{2D,j}$ (units $L^{-2}$) one obtains the collected 2D concentration $n_{2D,coll}$ (units $L^{-2}$). Since SIPs represent "2D" droplet
concentrations of $n_{2D,i} = \nu_i/\delta A$ and $n_j = \nu_{2D,j}/\delta A$, Eq. 24 follows. A collection can only occur, if a larger droplet (or SIP) $i$
overtakes a smaller droplet (or SIP) $j$. First, $z_i > z_j$ and $w_{sed,i} > w_{sed,j}$ must hold and second the overtake time $\Delta t_{OT} :=$
$(z_i - z_j) \times (w_{sed,i} - w_{sed,j})^{-1}$ must fulfill $\Delta t_{OT} \leq \delta t$. One can define the overtake probability $p^{OT}$ being 0 for $\Delta t_{OT} > \delta t$
and 1 for $\Delta t_{OT} \leq \delta t$, and the "2D" collection probability $p_{ij}^{WM2D} = K_{ij}^{WM2D}\delta A^{-1}$. Simulations will demonstrate that the
WM2D and WM3D formulations are statistically equivalent under certain conditions, i.e. $p^{OT} \times p_{WM2D}$ equals $p_{WM3D}$.
**2.3.3 Linear sampling variant**
The regular AON variant can be sped up by introducing a linear sampling technique (LinSamp) as done in Shima et al. (2009)
or Dziekan and Pawlowska (2017). $\lfloor N_{SIP}/2 \rfloor$ combinations of pairs $i - j$ are randomly picked, where each SIP appears exactly
in one pair (if $N_{SIP}$ is odd, one SIP is ignored). As only a subset of all possible combinations is numerically evaluated, the
extent of collisions is underestimated. To compensate for this, the probability $p_{crit}$ (or equivalently $\nu_{coll}$) is upscaled by a





scaling factor
$\gamma_{\text{corr}} = N_{\text{SIP}}(N_{\text{SIP}} - 1)/(2\lfloor N_{\text{SIP}}/2\rfloor)$         (25)
to guarantee an expectation value as desired. Clearly, this reduces the computational complexity of the algorithm from $O(N_{\text{SIP}}{}^2)$
to $O(N_{\text{SIP}})$. Multiple collections are more likely than in the regular quadratic implementation. The LinSamp variant becomes
the preferred choice if $N_{\text{SIP}}$ is large. If $\nu_{coll}$ is larger than both, $\nu_i$ and $\nu_j$, all AON versions as introduced so far would produce
negative weights. In order to prevent this, $\nu_{coll}$ is artificially reduced to $0.99\max(\nu_i, \nu_j)$ in such a case. This limiter is applied
in all AON implementations, but is particularly significant in the LinSamp version due to the upscaling of $p_{\text{crit}}$. Moreover, note
that LinSamp can be reasonably used only in conjunction with AON-WM3D, not AON-WM2D.
## 2.4 Boundary condition
At the lower boundary droplets leave the domain according to their fall speed. Using the LCM, the moment outflow $F_{l,\text{out}}$ is
determined by accumulating the contributions $\nu_p(\mu_p)^l$ of all SIPs $p$ that cross the lower boundary $z = 0\,\text{m}$. Due to the discrete-
ness of the crossings, instantaneous fluxes are actually averages of the past $200\,\text{s}$. Using the bin model, $F_{l,\text{out}}$ is diagnosed by
$$F_{l,\text{out}} = \sum_{p=1}^{N_{\text{BIN}}} g_{p,k=1}\,(\tilde{m}_{\text{bb},p})^{l-1}w_{sed}(\tilde{m}_{\text{bb},p})\frac{\ln 10}{3\,\kappa}. \qquad (26)$$
At the model top, the simplest condition is to have a zero influx. In this case, the column integrated droplet mass will decrease
once a non-zero flux across the lower boundary occurs. To realize a zero-influx condition in the Eulerian model, the mass
concentrations at the ghost cell level $nz+1$ are simply set to zero. In the Lagrangian model, a zero influx condition is naturally
implemented when no new SIP are created at the top of the column.
In both models, also a non-zero influx at the model top can be prescribed. One variant is to use periodic boundary conditions.
In the Lagrangian approach this is done by increasing the height $z_p$ of affected SIPs by $Lz$, once their height drops below $0$.
In the Eulerian model, $g_{p,nz+1}$ is identified with $g_{p,1}$. A second non-zero influx variant is a prescribed size distribution that
is advected into the domain with its respective fall speed. In the bin model, the prescribed DSD simply defines the $g_{i,nz+1}$-
values. In the Lagrangian model, new SIPs have to be introduced close to the model top. For this, a new SIP ensemble is drawn
from the prescribed DSD at each time step using the SingleSIP-init method. In order to place the SIPs in the column, it is
considered how far it would fall at most from the model top during one timestep: $z_\Delta(p) = w_{sed,p} \times \Delta t$. In a straightforward
implementation, one would create one SIP from each bin with a position $z_{new,p}$ uniformly drawn from $[Lz, Lz - z_\Delta(p)]$ and
weighting factor $\nu_{new,p} = \nu_p \times (z_\Delta(p)/\Delta z)$. This implementation has, however, several undesirable side-effects. For small,
slowly falling SIPs $z_\Delta(p)$ is much smaller than $\Delta z$. Applying this procedure in every time step leads to $\Delta z/z_\Delta(p)$ SIPs per GB
in the end. Hence, we refine this procedure by creating a SIP with probability $z_\Delta(p)/\Delta z$, a weighting factor $\nu_{new,p} = \nu_p$ and
$z_{new,p} \in [Lz, Lz - z_\Delta(p)]$. Note that if $z_\Delta(p)/\Delta z > 1$, then either $\lfloor z_\Delta(p)/\Delta z\rfloor$ or $\lceil z_\Delta(p)/\Delta z\rceil$ SIPs are created depending
on the probability $(z_\Delta(p)/\Delta z) - \lfloor z_\Delta(p)/\Delta z\rfloor$. This establishes a similar spatial SIP occurrence across the size spectrum with
one SIP per GB and bin on average. Moreover, SIP numbers do not scale any longer with $\Delta t$.





## 2.5 Terminology

Before we start discussing the results, we outline the terminology of the various model versions. On a first level, we differentiate between Eulerian (BIN) and Lagrangian approaches (LCM), which can be both applied in a box (0D) or column model (1D) framework. By default, BIN uses the MPDATA advection algorithm (clearly only in 1D) and Bott's collection algorithm. Alternatively, MPDATA can be replaced by the 1st order upstream scheme (US1) and Bott's collection algorithm by Wang's (Wang). The Lagrangian model versions differ only in the way AON is employed. By default, 3D well-mixedness (WM3D) is assumed and a quadratic sampling (QuadSamp) of the SIP combinations is used. Those simulations are also referred to as "regular". A second type of QuadSamp simulation assumes 2D well-mixedness (WM2D). Linear sampling of SIP combinations can be alternatively used for the WM3D-version. Accordingly, only the terms "regular", "WM2D" and "LinSamp" refer to a specific type of simulation, while "QuadSamp" and "WM3D" may denote options in several simulations ("QuadSamp" can be used with WM3D and WM2D, and "WM3D" can be used with QuadSamp and LinSamp).

By switching off sedimentation in the column model source code (as done in section 3.2), box model results are produced in each GB. In order to distinguish the latter simulations from AON box model results in U2017 they are refered to as "noSedi" (implicitly assuming WM3D).

## 3 Results

### 3.1 Validation exercises: pure sedimentation

Before we start comparing collection in column model applications, we highlight the differences introduced by the different numerical treatment of the sedimentation process. Two simple setups with an influx of an exponential DSD with $r_0 = 50\,\mu$m is prescribed. In the first case the domain is initially empty and fills over time (EmptyDom). In the second case, the upper half of the domain is filled and LWC and DNC decrease linearly to zero from the domain top to the domain middle (HalfDom). Fig. 4 shows the vertical profiles of normalised zeroth (left) and second (right) moments for EmptyDom (top) and HalfDom (bottom). Because of the lack of numerical diffusion, the solid LCM curves show the exact results, except for the error introduced by discretizing the influx DSD with a probabilistic approach. Each panel showcases a convincing agreement between the Eulerian and Lagrangian approach. Only the BIN-US1 solutions are slightly smeared out. The small wiggles in the LCM curves originate from the probabilistic influx condition. Even though the above agreement is favourable, it might be that the advection errors of differently sized droplets compensate each other in the Eulerian approaches. Hence in a second validation step, the computation of mass profiles is confined to certain droplet size ranges. Figure 5 shows such vertical profiles for EmptyDom. We see that for all four size ranges, the BIN results are smeared out relative to LCM. For the smallest size ranges both BIN versions are equally "bad" (top left panel). For the three remaining panels, the MPDATA curves (dashed) are closer to the LCM reference than the US1 curves (dotted). On the other hand, the MPDATA curves in the bottom right panel show some wiggles. Overall, the agreement between LCM and BIN-MPDATA is good. The discrepancies introduced by the different sedimentation treatment seem to be small enough to focus on the collection process in the following comparisons.



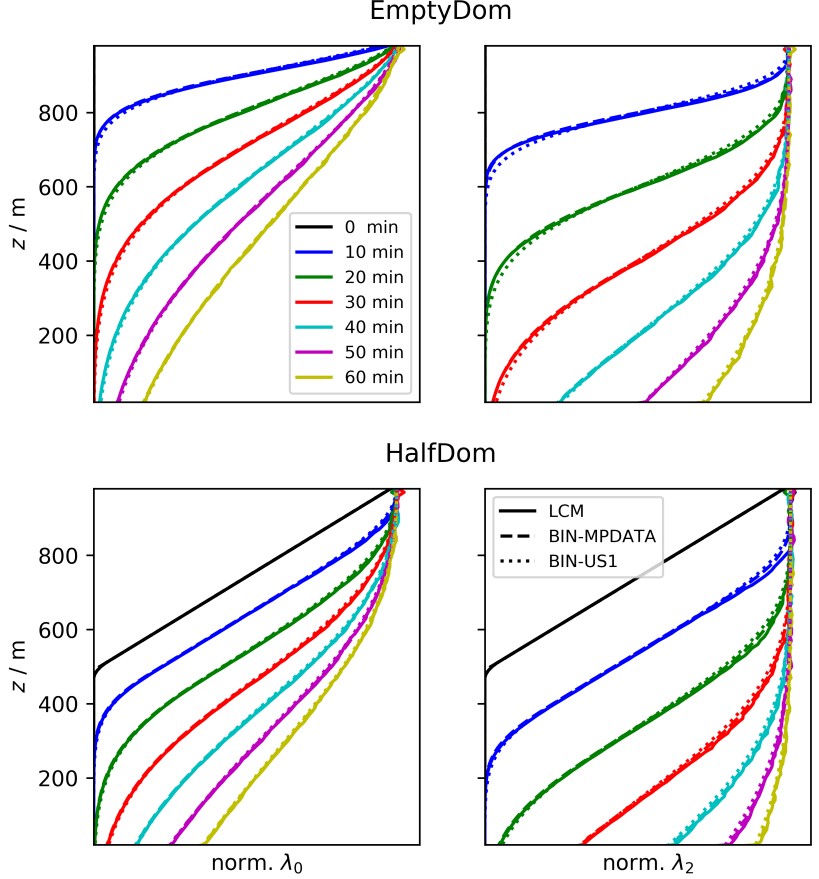

**Figure 4.** Pure Sedimentation test case: Comparison of BIN and LCM (solid) advection. BIN uses either MPDATA (dashed) or 1st order Upstream scheme (dotted). EmptyDom (upper row) and HalfDom (lower row) setup are used with an exponential distribution with $r_0 = 50\,\mu m$ as influx condition. Displayed are vertical profiles of normalised zeroth and second moment at the indicated points in time.

## 3.2 Box model emulation simulations

### 3.2.1 Regular AON version

In this section, we choose a column model setup that is supposed to produce results that are similar to box model results. For this, we initialise the default DSD in all GBs of the column and use periodic boundary conditions. In LCM1D, different SIP ensemble realisations of this DSD are initialised in each GB.

The deterministic bin column model predicts identical DSDs in all GBs, as in each GB the divergence of the sedimentation flux is zero. Hence, for this specific setup, the attained BIN1D results are identical to those of a corresponding BIN0D model or the data of Wang et al. (2007, see their Tables 3 and 4).



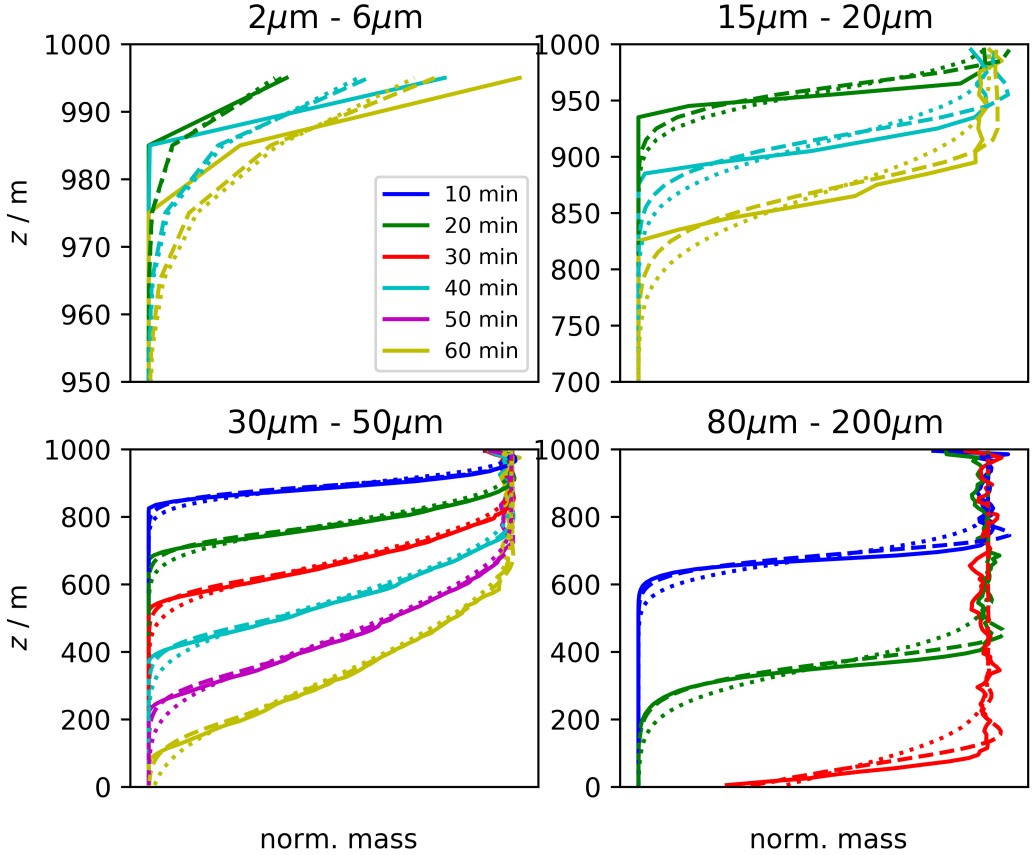

**Figure 5.** Pure Sedimentation test case: Comparison of BIN and LCM advection. EmptyDom setup with an exponential distribution with $r_0 = 50\,\mu$m as influx condition. Displayed are vertical profiles of normalised mass within specified size ranges (see on top of each panel) at the indicated points in time. Note that most panels use different $y$-axis ranges and do not show all six points in time.

In LCM1D, the combination of homogeneous initial conditions and periodic BCs results in statistically identical results
across all GBs. However, the averaged results may not be the same as in LCM0D, as lucky droplets/SIPs can collect other
droplets/SIPs not only from a single GB as in LCM0D, but from any GB (depending on how fast they fall), creating potentially
larger and/or faster growing lucky droplets/SIPs than in LCM0D. In other words, the number of SIPs interacting with each
other is increased in LCM1D. This, as we will show below, accelerates the convergence of the simulations.
Within the LCM1D-implementation, pure box model results can be obtained by switching off sedimentation. Without sed-
imentation, the GBs of the column are not interconnected and the collection process proceeds independently. In the follow-
ing, we refer to those simulations as "noSedi". By default, we use $nz = 50$ GBs with $\Delta z = 10\,$m (giving a column height of
$Lz = 500\,$m), $\Delta V = 1\,$m$^3, \Delta z = 10\,$m, $\Delta t = 10\,$s and $\kappa = 40$ throughout section 3.2. The results are averaged over $nr_{inst} = 20$
realisations. AON-WM3D is employed in LCM1D and sedimentation is switched on unless noted (for better discrimination



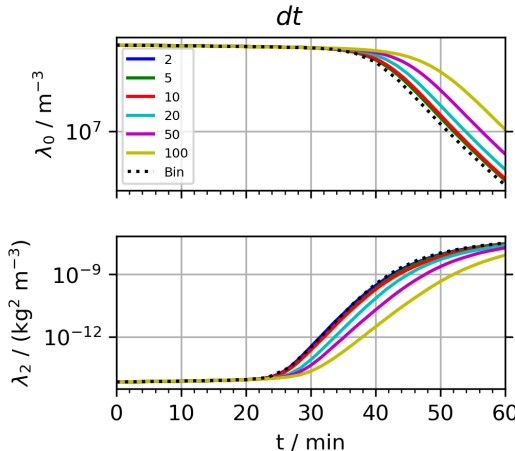

**Figure 6. BoxModelEmul setup:** Temporal evolution of column-averaged $DNC$ and $Z$ over one hour for various time steps $\Delta t$ (see inserted legend for $\Delta t$-values in seconds). All other parametes take the default values as given in the caption of Fig. 7.

from the noSedi, those simulations will be referred to as "full"). Moreover, the regular AON-WM3D version uses a quadratic sampling of SIP combinations (referred to as "QuadSamp").

Figure 6 shows the temporal evolution of column-averaged LCM1D moments $\lambda_l$ ($l = 0$ and 2) over one hour for various time steps $\Delta t$. The box model data serve as orientation in this and following Figures 6-9. We find that in terms of $\lambda_0$ and $\lambda_2$ LCM1D results converge for $\Delta t \leq 10\,\text{s}$. The noSedi simulations show a similar time step dependence (not shown). Hence, AON works surprisingly well for large time steps; a fact that was already shown with the AON box model (see Fig. 18 of U2017).

Next, we discuss the sensitivity to more physical and numerical parameters. We found that convergence is usually more easily reached for higher moments than for $\lambda_0$ (not shown). Hence in the following, we confine our analysis to the most "critical" quantity, and Fig. 7 displays the $\lambda_0$-evolution for various sensitivity experiments. Even though we analyse the results in some detail, we want to mention that the observed differences are in principle not substantial. In fact, results differ often much more due to a different collection kernel or slightly varied initial DSDs (see section 3.2.4). Nevertheless, the analysis will help to understand more deeply how collection works in an LCM with AON. This pronounced effort is justified, as precipitation initiation is still not fully understood and a well-validated Lagrangian approach may lead to new insights (Dziekan and Pawlowska, 2017; Grabowski et al., 2019).

In a first simple step, we vary $nz$ (see first row of Fig. 7), which changes two aspects of the numerical setup. The number of GBs over which interactions can occur and secondly the height of the column. This implicitly changes the time it takes for SIPs to fall through the total column and hence changes the "recycling" time scale $L_z/w_{sed}$. Together with $nz$, $nr_{inst}$ is varied such that $nz \times nr_{inst}$ is always 1000. Accordingly, all simulation results are averaged over the same number of GBs and we avoid that simulations with smaller $nz$ produce noisier data. In the noSedi-simulations (panel a), the moment evolution is not affected by varying ($nz$, $nr_{inst}$). This is trivial, as in any case the average is taken over 1000 independent GBs. At least, these results demonstrate that averaging over that many GBs suffices by far to produce robust averages. In the full simulations



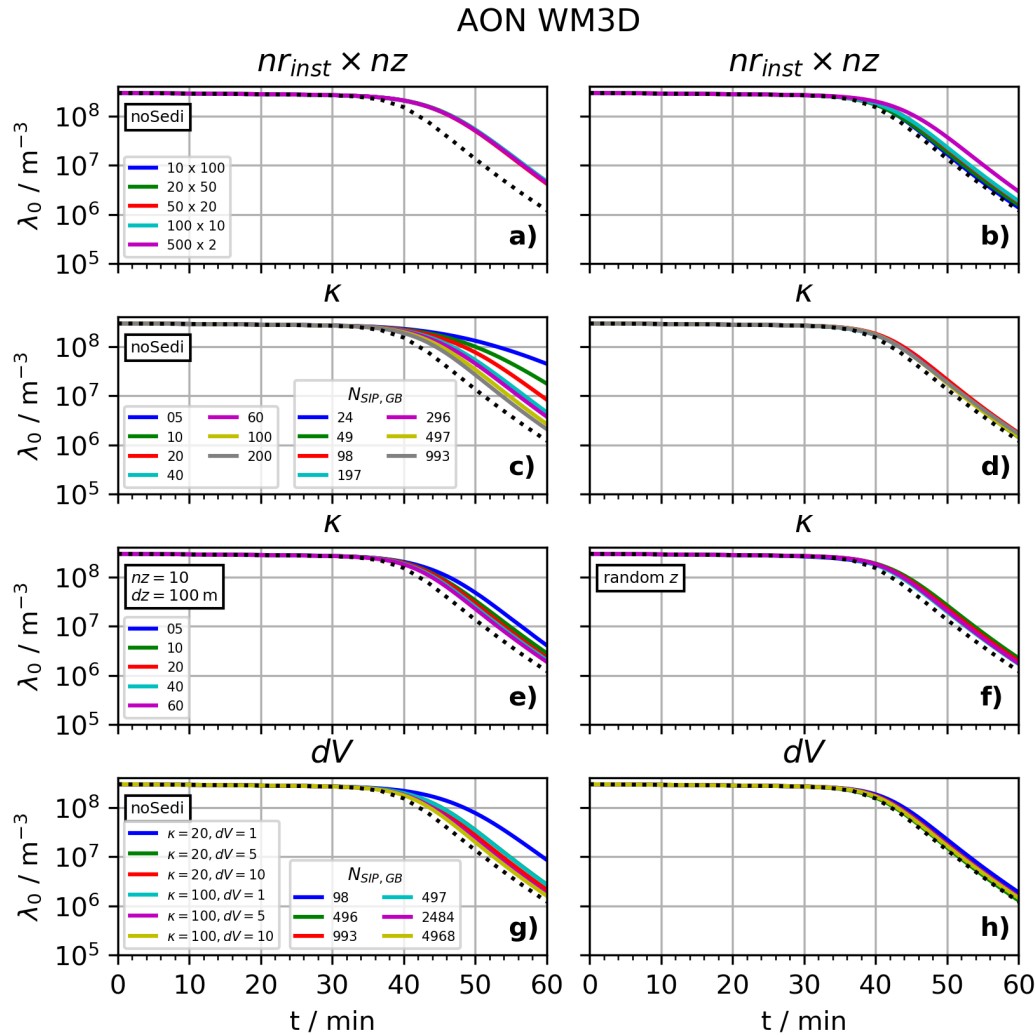

**Figure 7. BoxModelEmul setup:** Temporal evolution of column-averaged moment $\lambda_0$ (i.e. droplet concentration) over one hour. The default setting is $nz = 50, nr_{inst} = 20, \Delta V = 1\,\mathrm{m}^3, \Delta t = 10\,\mathrm{s}, \Delta z = 10\,\mathrm{m}, \kappa = 40$ and $L_z = nz \times \Delta z$. The microphysical parameters of the initial exponential droplet size distribution are $LWC = 1\mathrm{g/m}^3, r_0 = 9.3\,\mu\mathrm{m}$ and $DNC = 297\,\mathrm{cm}^{-3}$ as in many previous studies (Berry, 1967; Wang et al., 2007). The parameter or parameter pair that is varied is written on top of each panel and the legend lists the parameter values for the different colours. If further parameters (besides the varied parameter) take non-default values, it is indicated in a black box. In any case, the total number of GBs is $nr_{inst} \times nz = 1000$. By default, sedimentation is switched on. Simulations without sedimentation and independent rain formation in each GB (identical to a box model treatment) are labelled as "noSedi" (appear only in the left column).

1  (panel b), the $\lambda_0$-decrease is more pronounced and the various setups produce nearly identical results (except for the case with

2  $nz = 2$, which is in between the other full simulations and the noSedi simulations). From this finding alone one may argue that

3  the collection process is more efficient in LCM1D than in LCM0D.



The second row shows a variation of $\kappa$ which reveals qualitatively different convergence properties of the noSedi simulations (panel c) and the full simulations (panel d). In the noSedi simulations, an increase of $\kappa$ (and $N_{SIP}$; see extra legend for according $N_{SIP}$-values) leads to a faster decrease of $\lambda_0$. Large differences between $\kappa = 5$ and $40$ simulations are apparent; above $\kappa = 40$, an increase of $\kappa$ leads only to marginal improvements. Also for the highest $\kappa$, the $\lambda_0$-values remain slightly above the bin reference. For the smallest $\kappa$-value, only 24 SIPs are created according to Eq. 16 and interactions among that few computational particles overemphasize the impact of correlations. It is well-known that for small ensembles of real droplets correlations become important (Bayewitz et al., 1974; Wang et al., 2006). Analogously, we introduced correlations in our numerical approach by using too few computational particles. We believe that this hinders the formation of lucky droplets and fewer droplets get collected (hence $\lambda_0$ is larger for smaller $\kappa$). Another more technical explanation is that the $\nu_p$-distribution of the SIP ensemble is such that the formation of lucky SIPs is not supported. Ideally, there is a reservoir of SIPs with small $\nu$-values which can become lucky SIPs. There might be too few SIPs with small $\nu$ for small $\kappa$.

Contrarily, the full simulations (panel d) give nearly identical results independent of $\kappa$. We obtain converged results with as few as 24 SIPs in each GB. Compared to $\kappa = 200$ with 1000 SIPs, the simulations are a factor $40^2$ faster. The reason for the much faster convergence in terms of $N_{SIP,GB}$ is that the GBs are interconnected which effectively raises the number of potential collision partners. Drops with radius 100 and $500\,\mu m$ have fall speeds of around $0.7\,m\,s^{-1}$ and $4\,m\,s^{-1}$, respectively. Thus it takes them around $14\,s$ and $2.5\,s$ to fall through a $\Delta z = 10\,m$-GB and they enter a new GB every few time steps given $\Delta t = 10\,s$.

How strongly SIPs are interconnected across GBs in LCM1D should depend also on geometrical properties of the column. In the next setup, we investigate the $\kappa$-sensitivity in a column with $nz = 10$ and $\Delta z = 100\,m$ instead of $nz = 50$ and $\Delta z = 10\,m$ (panel e). Then, SIP interactions can occur only across 10 GBs and overall five times fewer SIPs are present in the column than for the default case with $nz = 50$. Moreover, the domain is stretched by increasing $\Delta z$ to $100\,m$, which increases the residence time of a SIP in a GB by a factor 10, slowing down additionally SIP interactions across GBs. Those two changes introduce a weak $\kappa$-dependence, yet much weaker than in the corresponding noSedi-simulations (panel c).

In a technical experiment, sedimentation is turned off, but SIPs are randomly redistributed inside the column after each time step (panel f) similar to Schwenkel et al. (2018). Again, we find converged results for small $\kappa$-values down to 5 (panel f). This elucidates that convergence is improved once some process exchanges SIPs between GBs, may it be for physical reasons like sedimentation or by an artificial operation as the randomized SIP re-location. We speculate that in full 2D/3D LCM-simulations turbulent motions and sedimentation increase the SIP exchange across GBs and hence may additionally increase the performance of AON. The two latter simulation series are promising, as they suggest that in a column model (and probably also 2D/3D model) convergence is potentially reached with fewer SIPs per GB than in a box model. Nevertheless the tests also highlight that convergence with $\kappa$ depends on many circumstances and convergence tests are prerequisite to any LCM simulation with AON.

In bin models, the Smoluchowski equation, which is strictly valid only for an infinite volume and hence an infinite number of well-mixed droplets, is solved. Accordingly, only concentrations are prescribed in bin model algorithms. Neither $\Delta V$ nor the absolute number of droplets is considered in this approach. At least in the limit of all SIPs having weighting factor $\nu = 1$,





the AON algorithm solves the master equation (Dziekan and Pawlowska, 2017) which takes into account $\Delta V$ and results may
depend on the actual number of involved droplets. Clearly, correlations (which are accounted for in the master equation) are
larger in smaller volumes (Bayewitz et al., 1974; Wang et al., 2006; Alfonso and Raga, 2017).
For the given SIP-initialisation procedure, $N_{SIP,GB}$ depends solely on the chosen $\kappa$-values and is independent of $\Delta V$. By
construction, a $\Delta V$-variation does not affect at all the simulation results, as all SIP weights are simply rescaled. Indeed, we
obtain nearly bit-identical results for a $\Delta V$-variation. To explore the $\Delta V$-sensitivity in our LCM1D, the SIP-init procedure has
to be adapted. In the adapted version the SIP number increases proportionally with $\Delta V$ as it would in reality. As computational
requirements increase quadratically with $N_{SIP,GB}$, the variation of $\Delta V$ and $N_{SIP,GB}$ can be performed only for a small range
of $\Delta V$-values. $\Delta V$ is increased by a factor of five or ten. As a base case, we use the simulations with $\kappa = 20$ and $\kappa = 100$
and define $\Delta V := 1\,\mathrm{m}^3$. The fourth row shows results for the noSedi (panel g) and the full simulations (panel h). Apparently,
the noSedi-simulations with larger $\Delta V$ converge to the solution we obtained before by using a sufficiently large $\kappa$. In full
simulations, a $\Delta V$-variation has basically no effect. The $\kappa = 100, \Delta V = 10\,\mathrm{m}^3$-simulation considered on average collisions
between 5000 SIPs in each GB. Yet, the results are basically identical to the case $\kappa = 5, \Delta V = 1\,\mathrm{m}^3$ with 24 SIPs in each GB
(which runs nearly 40000 times faster).
In the present simulations where SIPs with weights $\nu > 1$ are used, variations of the numerical parameter $\kappa$ and the physical
parameter $\Delta V$ are interconnected and their effects cannot be disentangled. Hence, the AON algorithm can only answer whether
correlations matter in systems with a certain number of SIPs. These correlations are not necessarily the correlations one would
see in a real system with millions to billions of real droplets. Nevertheless, the last sensitivity series implies that at least in our
model system the importance of correlations are likely the same in a system with $N_{SIP,GB} = 24$ and with $N_{SIP,GB} \approx 5000$.
Assuming that the importance of correlations in a real system with billions of droplets is similar to that of a system with 5000
SIPs, the latter finding demonstrates that LCMs can capture the collection process with astonishingly few SIPs.
The noSedi $\kappa$-sensitivity series as shown in panel c) was already presented in Fig. 18 of U2017. There it was found that for
high enough $\kappa$ the LCM0D results lie below the BIN0D reference contradictory to the present noSedi simulations. The reason
for this inconsistency is a programming bug in the LCM0D-AON version used in U2017. The Hall/Long kernel values are
stored in look-up tables and were wrongly accessed (overestimating the actual mass of the involved droplets by 2%). Hence,
the collection process proceeded more rapidly in U2017. Despite this flaw, the main findings of U2017 remain valid. Yet the
more rapid collection of LCM0D-AON in U2017 should clearly not be attributed to conceptual differences of AON and BIN
algorithms.
### 3.2.2  AON with linear sampling
Figure 8 displays again the $\lambda_0$-evolution in $\Delta t$- and $\kappa$-sensitivity studies, now f or the WM3D version with linear sampling
(LinSamp). The left/right column of the figure shows results without/with sedimentation. For the default time step of $\Delta t = 10\,\mathrm{s}$,
results do not converge and are far off the desired result (first row). Reducing the time step to $\Delta t = 1\,\mathrm{s}$ increases the number of
tested collisions by a factor of 10. This seems to be a crucial point as the results now converge (second row); for the noSedi-case
only for the highest $\kappa$-values, for the full simulation for any $\kappa$.



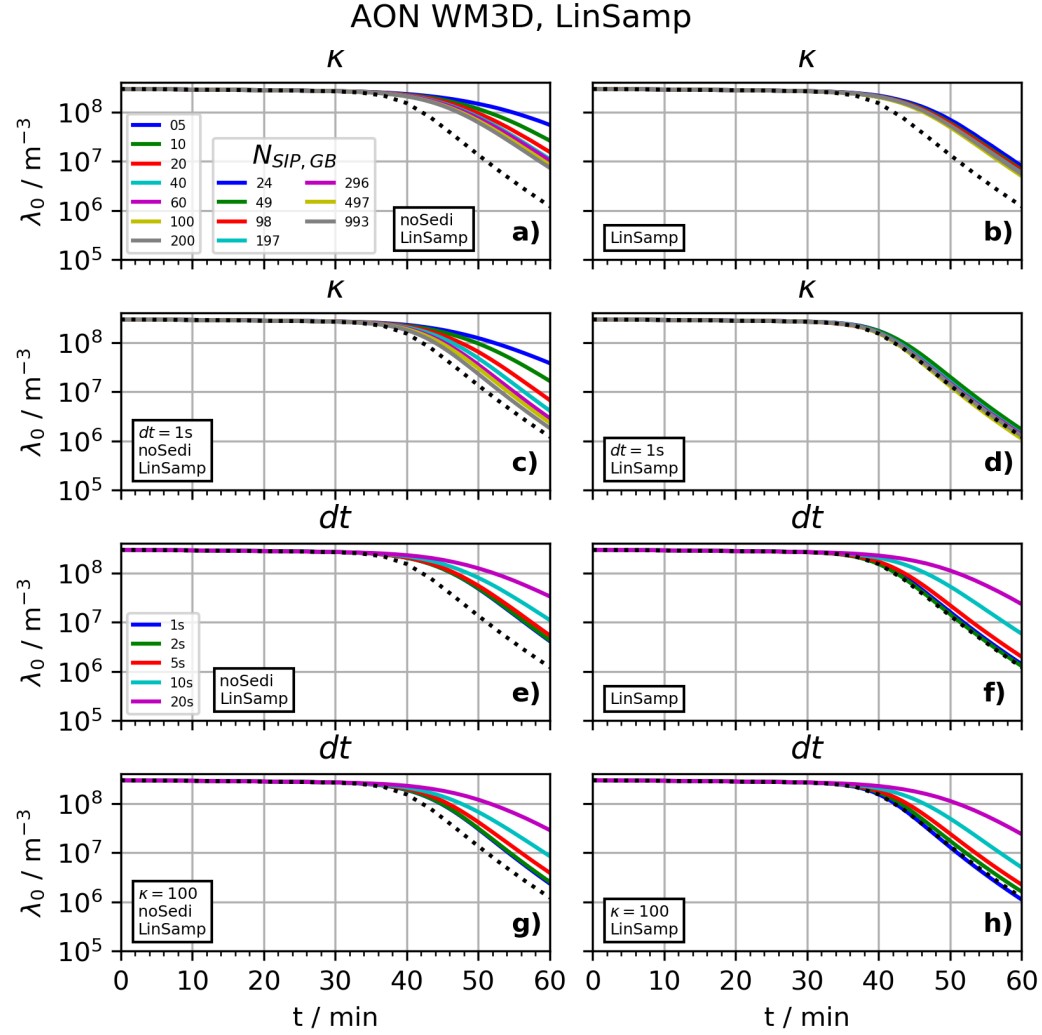

**Figure 8. BoxModelEmul setup:** The plots are analogous to Fig. 7 (all setup parameters are listed in that caption), now simulations with linear sampling (as described in section 2.3.3) are depicted. The left column shows noSedi simulations, the right column shows LCM1D simulations..

Finally, $\Delta t$ is varied between 1 and $20\,\mathrm{s}$. This is roughly the $\Delta t$-range for which the QuadSamp simulations produced
more or less converged results. Here, we find convergence only for time steps as small as $5\,\mathrm{s}$. We attribute this "delayed" $\Delta t$-
convergence to the fact that SIP combinations, where $\nu_{coll}$ is limited to $0.99\max(\nu_i, \nu_j)$, occur too often and that this "limiter"
effect becomes negligible only for small enough time steps.
In general, we find that switching off sedimentation in the LinSamp simulations deteriorates the convergence properties, as
already seen in the QuadSamp simulations.

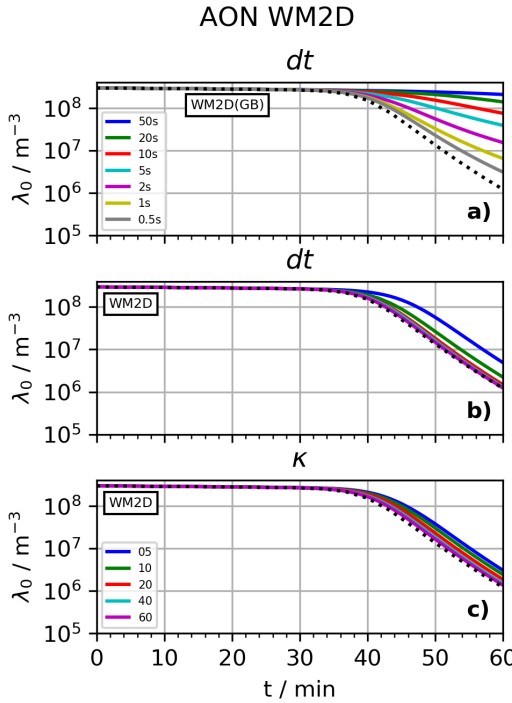

**Figure 9. BoxModelEmul setup:** The plots are analogous to Fig. 7 (all setup parameters are listed in that caption), now simulations with explicit overtakes and a 2D well-mixed assumption (as described in section 2.3.2) are depicted. In the top panel overtakes are considered only between SIPs inside the same GB, whereas the other panels show the regular version where overtakes are tested for all SIPs of the column.

.

All in all, convergence in the LinSamp simulations is reached only for smaller $\Delta t$ relative to the QuadSamp simulations. In
Hence, the potential benefit of the reduced computational cost may be outweighed by the stronger requirements on $\Delta t$. In
particular, in full 2D/3D LCMs also condensation/deposition and sedimentation has then to be solved more often unless subcy-
cling is introduced. Whether LinSamp or QuadSamp is in the end more efficient in a full 2D/3D LCM may depend also on the
simulated cloud type and the complexity of the LCM (inclusion of aerosol physics, chemistry or different hydrometeor types,
e.g. as in Jaruga and Pawlowska, 2018; Brdar and Seifert, 2018). And indeed, Dziekan et al. (2019) presents 2D and 3D LCM
simulations using the LinSamp approach and they see convergence only for a rather small time step of $dt = 0.1\,\mathrm{s}$, which is
probably caused by the slow convergence of LinSamp.
**3.2.3   AON version with explicit overtakes**
Next, we will discuss results of the AON-WM2D version with explicit overtakes. Figure 9 displays again the temporal evolution
of $\lambda_0$. For the chosen setup with homogeneous initial conditions and periodic boundary conditions, 3D well-mixedness of the
SIPs is expected to be maintained over the course of the simulation. Hence, the AON-WM3D and AON-WM2D version are



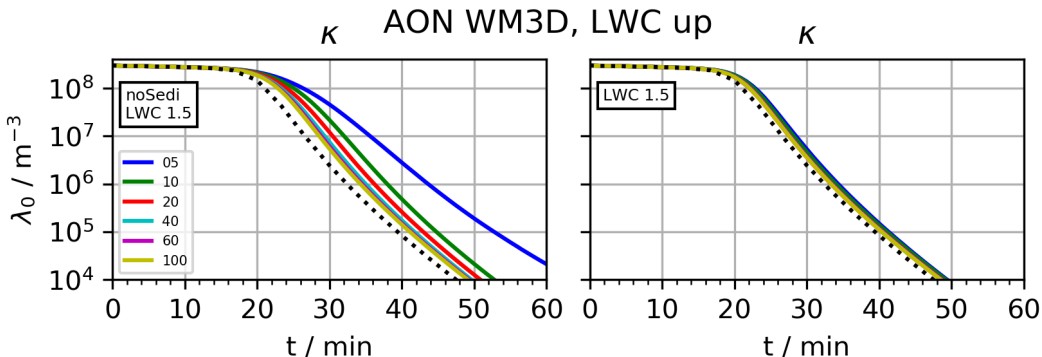

**Figure 10. BoxModelEmul setup:** The plots are analogous to Fig. 7 (all setup parameters are listed in that caption) and the sensitivity to $\kappa$ is depicted for simulations with initial $LWC = 1.5 \, \text{g/m}^3$. The left and right panel juxtapose noSedi and full simulations.

supposed to produce similar outcomes. Panel a shows results for the version where only intra-GB overtakes are considered.
Results are far off the benchmark curve, only for the smallest time step of $\Delta t = 0.5 \, \text{s}$ they tend to approach the reference. Panel
b shows the same $\Delta t$-variation (down to $\Delta t = 2 \, \text{s}$) for the version where overtakes are considered across the full column. In
the present example, it was also necessary to check for overtakes across the periodic boundary. Then, convergence is reached
for $\Delta t \leq 10 \, \text{s}$, very similar to the regular AON-WM3D version. The bottom row shows a slight dependence on $\kappa$, yet AON
WM2D results seem to converge to the WM3D-results.
Overall, we can conclude that the feasibility and correct implementation of the WM2D-variant was demonstrated, with the
caveat that overtakes have to be considered in the full column.
**3.2.4  Microphysical and bin model sensitivities**
So far, all simulations were initialised with the same initial DSD, the same collection kernel, and the results are compared to
the same bin reference. Accordingly, in this section, we perform simulations with modified $LWC, r_0$ and $DNC$. Moreover,
we highlight the effect of the employed kernel on the AON performance. And finally, we also present bin model sensitivities
(namely, we switch from Bott's algorithm to Wang's algorithm and vary the bin resolution and the time step).
In a first experiment, we increase $LWC$ by a factor of 1.5 and do again a $\kappa$-sensitivity test (Fig. 10). We keep DNC fixed
and hence the mean radius is $r_0 = 9.3 \, \mu\text{m} \times 1.5^{(1/3)} = 10.7 \, \mu\text{m}$. Compared to the base case with $LWC = 1 \text{g/m}^3$, $\lambda_0$ starts to
decrease after 20 minutes (instead of 40 min) and $\lambda_0$ decreases below $10^4 \, \text{cm}^{-3}$ (instead of $10^6 \, \text{cm}^{-3}$). In the full simulations
(right panel), we again find results nearly independent of $\kappa$. In the noSedi-sims (left panel), fewer SIPs are necessary to obtain
reasonable results compared to the base case (see Fig 7c).
In a next step, the characteristics of the initial DSD are more flexibly varied for fixed $\kappa = 40$. For such a $\kappa$-value the noSedi-
simulation of the base case was considerably off the reference. Figure 11 shows the temporal evolution of the mean diameter,
$\lambda_0$ and $\lambda_2$ (from top to bottom) over $100 \, \text{min}$. Simulations with the Bott model are contrasted with the regular AON-WM3D,



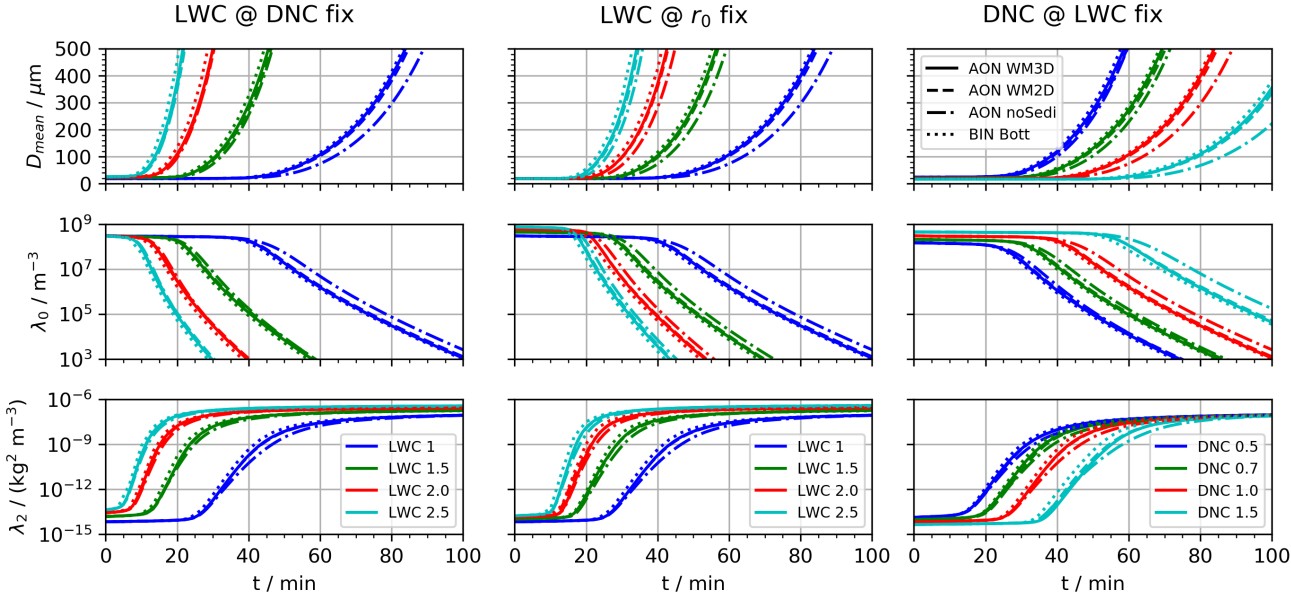

**Figure 11. BoxModelEmul setup:** Figure analogous to Fig. 7 (all setup parameters are listed in that caption), now displaying also the temporal evolution of the mean diameter (top row) and the second moment $\lambda_2$ (bottom row) additional to $\lambda_0$ (middle row). Variations of the initial size distribution parameters $LWC = \lambda_1(t=0), r_0$ and $DNC = \lambda_0(t=0)$ are performed. The first and second column show a variation of $LWC$ (see inserted legend) for either fixed $DNC$ or $r_0$. The third column shows a $DNC$-variation for fixed $LWC$. Four different models are used (AON-WM3D, AON-WM2D, noSedi and BIN1D; see legend in top right panel).

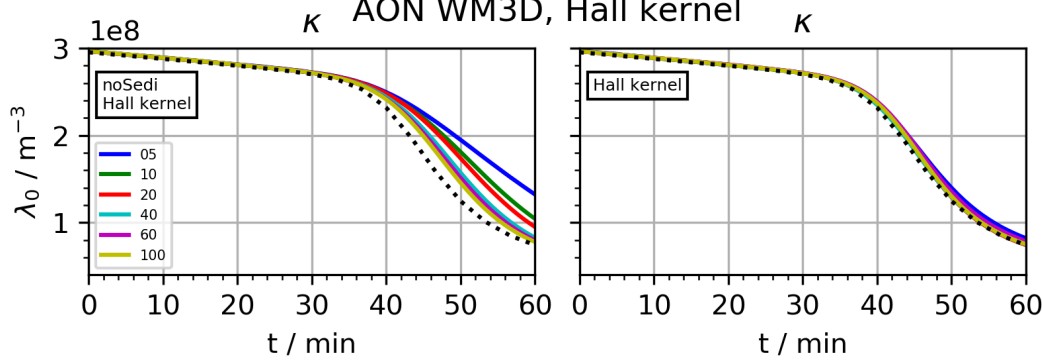

**Figure 12. BoxModelEmul setup:** The plots are analogous to Fig. 7 (all setup parameters are listed in that caption) and the sensitivity to $\kappa$ is shown for simulations with the Hall kernel. The left and right panel juxtapose noSedi and full simulations. Unlike to previous plots, the y-axis uses a linear scale.

1    AON-WM2D and AON-noSedi. The first two columns show simulations for a variation of the initial $LWC_0 = \lambda_1(t_0)$, for

2    either fixed droplet number or fixed mean radius. The right-most column shows a variation of the initial droplet number. The



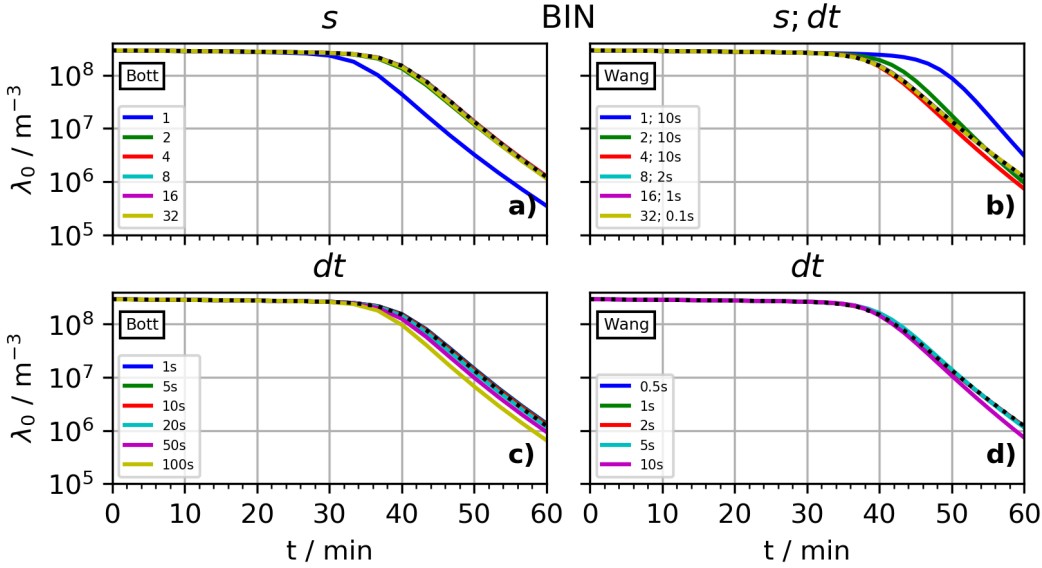

**Figure 13. BoxModelEmul setup:** The plots are analogous to Fig. 7. The left and right panel juxtapose BIN results with Bott's and Wang's algorithms. The default parameters are $s = 4$ and $dt = 10s$. Unlike to the AON case, the choice of $nz$ is irrelevant.

default value (denoted as "1" in the legend) is scaled by factor of $1.5, 2.0$ or $2.5$ (for a LWC-variation) and $0.5, 0.7$ or $1.5$ (for
a DNC-variation). We find for most cases, that the three model versions produce very similar $\lambda_2$-evolutions. The bin model
predicts in all cases slightly higher droplet numbers $\lambda_0$ than the AON version. The WM2D are in between the WM3D and the
bin model. As a consequence, the mean droplet diameter increases the fastest with the WM3D version.
Figure 12 shows simulations where the Long kernel is replaced by the Hall kernel. The decrease in DNC occurs at a slower
rate (the y-scale now uses a linear scale). For the full simulations (right), we obtain perfect agreement for any chosen $\kappa$-value.
Moreover, convergence with $\kappa$ in the noSedi-simulations (left) is less critical than in the base case and results converge for
$\kappa \geq 40$.
We conclude the box model emulation section by showing sensitivities of the bin model approach. For this, we vary the bin
resolution $s$ and the time step for the base case with $LWC = 1\text{g/m}^3$ and Long kernel. The default time step is again $dt = 10\,\text{s}$
and the bin resolution is $s = 4$. The left and right column of Fig. 13 show results obtained with Bott's and Wang's algorithm,
respectively. The black reference curve in Figs. 6 to 9 are data from Wang's algorithm with $s = 16$ and $dt = 1\,\text{s}$ and is also added
to the present plot for orientation. We find that Bott's algorithm converges for $s \geq 2$. For higher resolutions, Wang's algorithm
does not produce stable results for $dt \geq 10\,\text{s}$ and the time step had to be reduced (see inserted legend, for the combination of
$s$ and $dt$). For $s \geq 8$ results have converged to the reference. The second row shows the time step dependency for a medium
resolution of $s = 4$. Bott's results are reliable for $dt$ as high as $100\,\text{s}$ and converge for $dt \leq 20\,\text{s}$. On the other hand, Wang's
algorithm requires $dt \leq 10\,\text{s}$ and convergence is reached for $dt \leq 5\,\text{s}$. Overall, we can conclude that both algorithms converge
to the same values, given a sufficiently high $s$ and low $dt$ is chosen. As Bott's algorithm seems to be more robust than Wang's





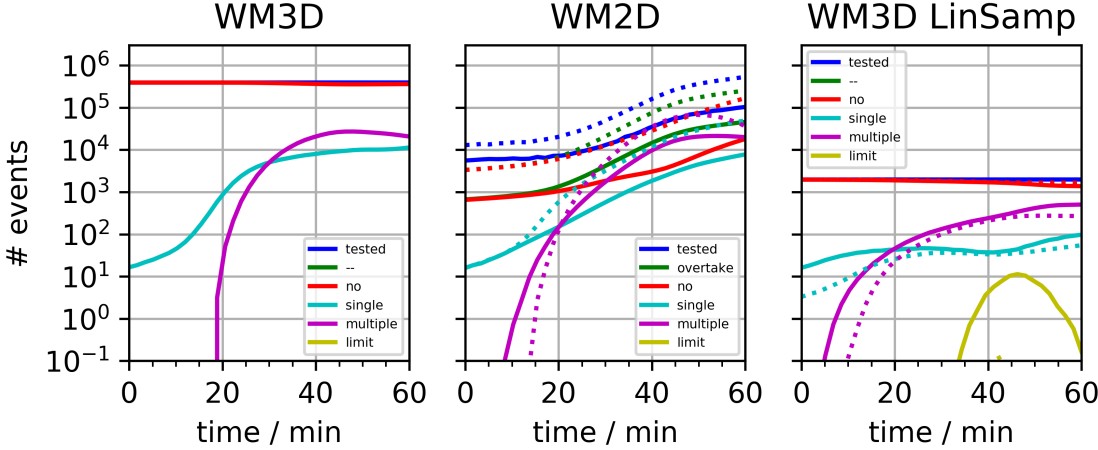

**Figure 14. BoxModelEmul setup:** Time series of number of events in the various AON implementation. Shown are the number of tested SIP combinations, of overtakes, of no collection, of a single collection, of a multiple collection in every time step. Additionally, the number of limiter cases, where $n_{coll}$ had to be artificially reduced, is shown (occurs only in the LinSamp-panel). The parameter setup is given in the text. In the WM2D-panel, the dotted lines show the case with $dz = 10\,\text{m}$. In the LinSamp-panel, the dotted lines show the $1\,\text{s}$-simulation. The displayed numbers can be below unity, as averages over 20 instances are shown.

algorithm, all following bin model simulations are carried out with this algorithm. Comparing the various collection algorithms,
we find that Bott's algorithm has the least requirements in terms of bin resolution and time step as we have converged results
for $t$ up to $100\,\text{s}$ and $s$ as low as 2. AON simulations may converge for $\kappa = 5$ (corresponds roughly to $s = 2$) and $dt = 10\,\text{s}$ if
GBs of the column are sufficiently interconnected and averaging over several realisations is done. Wang's algorithm produces
correct solutions for $s = 4$ and $dt = 5\,\text{s}$, yet increasing the bin resolution has to be done hand in hand with a reduction of the
time step.
**3.3 Analysis of the "algorithmic interior"**
Now, we turn the attention to the processes in the "algorithmic interior" of the various AON versions. Figure 14 and Tab. 2 give
an example of how often collections occur in the model. For AON-WM2D, also the number of overtakes is given. The listed
numbers give a rough indication of the importance of the various events (overtake, no collection, single collection, multiple
collection, limiter), yet we want to note the caveat that the relative importance changes with a change of the parameter setup.
Here, results are shown for the specific setup with $nz = 20, nr_{inst} = 10, \Delta V = 1\,\text{m}^3, \Delta t = 5\,\text{s}, \Delta z = 50\,\text{m}$ and $\kappa = 40$. The
figure shows qualitatively the number of occurences as a function of time, whereas the table gives aggregate values for three
20 min blocks and the total 60 min simulation period. In both WM3D versions (regular QuadSamp and LinSamp), the number
of tested SIP combinations $N_{\text{comb}}$ is constant over time. Clearly, the LinSamp value is smaller by a factor of 200 ($= N_{\text{SIP}}$).
For the WM2D-approach, on the other hand, $N_{\text{comb}}$ increases over time as the DSD gets more mature and larger droplets fall




**Table 2. BoxModelEmul setup:** Number of events for various AON model variants for the parameter setup given in the text. $N_{comb}$ is the number of tested SIP combinations and $N_{LI}$ is the number of limiter cases, where $n_{coll}$ had to be artificially reduced. Moreover, $\eta_{OT}, \eta_{NO}, \eta_{SI}$ and $\eta_{MU}$ specify the number of overtakes, no collections, single collections and multiple collections divided by $N_{comb}$. The two last columns shows summed up $p_{\text{crit}}$ (summed over all times and SIP combinations/overtakes) and the average $p_{\text{crit}}$. For each model variant, the first three rows show aggregate values over three time periods ($0-20\,\text{min}, 20-40\,\text{min}$ and $40-60\,\text{min}$) and the fourth row values for the full time period.

| Model variant | tested SIP combinations $N_{comb}$ | overtakes $\eta_{OT}$ | no collection $\eta_{NO}$ | single collection $\eta_{SI}$ | multiple collection $\eta_{MU}$ | limiter event $N_{LI}$ | $\sum p_{\text{crit}}$ | $\bar{p}_{\text{crit}}$ |
|---|---|---|---|---|---|---|---|---|
| | 9.44e7 | - | 100.0% | 0.0% | 0.0% | 0 | 2.91e4 | 3.08e-4 |
| block #1 | 9.44e7 | - | 97.0% | 1.2% | 1.8% | 0 | 4.25e7 | 4.50e-1 |
| AON WM3D | 9.45e7 | - | 91.2% | 2.5% | 6.3% | 0 | 1.95e8 | 2.06e0 |
| | 2.83e8 | - | 96.1% | 1.3% | 2.7% | 0 | 2.38e8 | 8.38e-1 |
| | 1.49e6 | 13.9% | 12.7% | 0.8% | 0.3% | 0 | 2.70e4 | 1.30e-1 |
| block #2 | 3.83e6 | 34.7% | 11.9% | 4.5% | 17.8% | 0 | 3.64e7 | 2.74e1 |
| AON WM2D | 1.77e7 | 44.1% | 12.1% | 6.4% | 25.3% | 0 | 2.15e8 | 2.75e1 |
| | 2.30e7 | 40.6% | 12.2% | 5.8% | 22.5% | 0 | 2.52e8 | 2.69e1 |
| | 3.64e6 | 28.6% | 27.7% | 0.7% | 0.0% | 0 | 2.85e4 | 2.74e-2 |
| block #3 | 1.53e7 | 43.9% | 22.0% | 6.5% | 14.9% | 0 | 3.62e7 | 5.37e0 |
| AON WM2D, $dz = 10\,\text{m}$ | 8.89e7 | 47.5% | 23.9% | 8.4% | 15.0% | 0 | 1.79e8 | 4.24e0 |
| | 1.08e8 | 46.4% | 23.8% | 7.9% | 14.5% | 0 | 2.15e8 | 4.31e0 |
| | 4.76e5 | - | 98.0% | 1.6% | 0.5% | 0 | 2.89e4 | 6.07e-2 |
| block #4 | 4.76e5 | - | 90.9% | 2.2% | 6.9% | 122 | 3.48e7 | 7.32e1 |
| AON WM3D, LS | 4.76e5 | - | 76.3% | 3.2% | 20.5% | 1343 | 3.21e8 | 6.75e2 |
| | 1.43e6 | - | 88.4% | 2.3% | 9.3% | 1465 | 3.56e8 | 2.49e2 |
| | 2.38e6 | - | 99.3% | 0.6% | 0.1% | 0 | 3.31e4 | 1.39e-2 |
| block #5 | 2.38e6 | - | 93.0% | 1.7% | 5.3% | 14 | 4.45e7 | 1.87e1 |
| AON WM3D, LS, $dt = 1\,\text{s}$ | 2.38e6 | - | 84.6% | 2.1% | 13.3% | 24 | 2.14e8 | 8.99e1 |
| | 7.14e6 | - | 92.3% | 1.5% | 6.2% | 38 | 2.58e8 | 3.62e1 |

faster. Relative to the regular WM3D, $N_{\text{comb}}$ of WM2D is at any time smaller. In the beginning of the simulation, possible
overtakes occur among relatively few SIPs; much fewer on average than there are in a GB, hence the total $N_{\text{comb}}$ is around a
factor 60 smaller (in the first 20 minutes; $9.44 \cdot 10^7$ vs. $1.49 \cdot 10^6$). Even towards the end of the simulation, many SIPs are still
small and travel through a small fraction of the GB. Only few SIPs grow to rain drop size and travel distances of order $\Delta z$. The
table shows that the total (time-integrated) $N_{\text{comb}}$ is more than a factor 12 smaller for WM2D than for WM3D ($2.30 \cdot 10^7$ vs.



$2.83 \cdot 10^8$). This demonstrates the numerical efficiency of the current WM2D implementation despite a theoretically unfavorable
computational complexity with a factor $nz$ higher $N_{\text{comb}}$ compared to the regular WM3D version. Moreover, the workload per
time step is constant in the WM3D-versions and determined solely by $N_{\text{SIP}}$. In the WM2D-version, the workload depends
additionally on the properties of the DSD and also on $\Delta z$. If $\Delta z$ is reduced by a factor of 5 (see block #3 in the table), $N_{\text{comb}}$
roughly increases by the same factor.
In the table, the ratios $\eta_{NO}, \eta_{SI}$ and $\eta_{MU}$ (find their definitions in the caption of the table) add up to $100\%$ for WM3D
(QuadSamp and LinSamp). In the regular WM3D version, only $1.3\%$ and $2.7\%$ of all tested combination lead to a single
or multiple collection. So, for most combinations $p_{\text{crit}}$ is close to zero and makes a collection unlikely. On the other hand,
for favourable SIP combinations $p_{\text{crit}}$ can be far above 1 (imagine a SIP combination with $\nu_i = 10^6, \nu_j = 10^2$ and $\nu_{coll} = 10^4$
yielding $p_{\text{crit}} = 100$). This also explains the somewhat surprising fact that the average $\bar{p}_{\text{crit}}$ is close to unity ($= 0.83$, see right-
most column). The PDF (probability density function) of all $p_{\text{crit}}$-values is strongly right-skewed (not shown). In the LinSamp
case, single and multiple collections occur in $2.3\%$ and $9.3\%$ of the tested combinations. Collections are more likely as $\bar{p}_{\text{crit}}$
is larger due to the upscaling. Moreover, $\nu_{coll}$ had to be artificially reduced in $N_{LI} \approx 1400$ cases. Note that such limiter cases
do not appear in the QuadSamp simulations. In the LinSamp version, $N_{LI}$ can be cut down by choosing a smaller time step
(see fourth block in table). Using $dt = 1\,\text{s}$ leads to 5 times smaller $p_{\text{crit}}$-values, increases $\eta_{NO}$, and decreases $\eta_{SI}$ and $\eta_{MU}$.
Limiter cases appear only in 38 of all combinations. For clarification, $p_{\text{crit}}$ of a single SIP combination scales with $dt^{-1}$; from
this, however, does not follow that the listed $\bar{p}_{\text{crit}}$-values of the two LinSamp simulation differ by a factor of 10, as the DSDs
and SIP ensembles/weights evolve differently in the two simulations.
Finally, we focus on the WM2D-version (block #2). Here, the sum of $\eta_{NO}, \eta_{SI}$ and $\eta_{MU}$ yields $\eta_{OT}$, and not $100\%$ as
before. In the end, around $40\%$ of all tested SIP combinations undergo an overtake. This quite large fraction comes from the
fact that the DSD (or more precisely the size distribution of the SIPs) features a strong bimodal spectrum. So most tested
combinations are combinations between a large collector SIP $i$ and a small SIP $j$ with $z_i > z_j$. Tested SIP combinations fulfill
by design $z_i(t + \Delta t) < z_j(t)$. For small SIPs $j$, $z_j(t + \Delta t) = z_j(t + \Delta t) - \epsilon$ holds. As $\epsilon$ is a small distance, it is likely that
$z_i(t + \Delta t) < z_j(t + \Delta t)$ is fulfilled, i.e. SIP $i$ overtakes SIP $j$. In more than every second overtake, a multiple collection occurs
(i.e. $\eta_{MU}/\eta_{OT} = 0.56$). In one eights/one third of the overtakes a single/no collection happens. So the relative importance of
the various events is quite different compared to the regular AON and also $\bar{p}_{\text{crit}}$ is three times larger (2.69 vs. 0.83). Note that
Changing $dz$ in the WM2D-simulation (block #3) also affects the relative occurences of no/single/multiple collections.
In all five setups we find, that in the end more multiple collections than single collections appeared. Except for the LinSamp
version with $dt = 10\,\text{s}$, the simulations converge. Clearly, the occurence of multiple collections in a simulation does not nec-
essarily deteriorate the simulation results. It is certainly not the case, that the time step choice or adaptation must be such that
multiple collections barely appear in a simulation. The present analysis only shows a correlation between the appearance of
limiter cases and a non-converged simulation. Strictly speaking, we cannot even say that the limiter cases are the reason for the
failure.
Several of the above findings may hold only for the specific setup used here. To put the findings into a broader context,
we next derive scaling relations for basic numerical quantities and, in particular, discuss their sensitivity to the time step and



the number of SIPs. For a simplified presentation, we limit ourselves to the WM3D versions with QuadSamp and LinSamp
and assumed converged simulation results and no limiter events. Moreover, we assume that an increase of $N_{SIP}$ leads to an
uniform decrease of all SIP weights $\nu_p$.
For the following basic quantities we have

$$\nu_p \propto \frac{1}{N_{SIP}}; \; nt \propto \frac{1}{\delta t}; \; N_{combs} \propto N_{SIP}{}^\alpha; \; \gamma_{\text{corr}} \propto N_{SIP}{}^\beta, \tag{27}$$

where $\gamma_{\text{corr}}$ is the correction factor defined in Eq. 25. For QuadSamp $\alpha = 2, \beta = 0$ and for LinSamp $\alpha = 1, \beta = 1$.
Accordingly,

$$\nu_{coll} \propto \frac{1}{N_{SIP}{}^2} \times \delta t, \tag{28a}$$

$$\nu_{sum} := \sum^{nt, N_{combs}} (\nu_{coll} \, \gamma_{\text{corr}}) \propto \frac{N_{SIP}{}^{\alpha+\beta}}{N_{SIP}{}^2} = 1, \text{ and} \tag{28b}$$

$$\bar{p}_{crit} := \frac{1}{N_{combs} \, nt} \sum^{nt, N_{combs}} (\nu_{coll}/\nu_p \, \gamma_{\text{corr}}) \propto N_{SIP}{}^{\beta-1} \, \delta t. \tag{28c}$$

In all versions $\nu_{sum}$ is independent of $N_{SIP}$ and $\delta t$. Clearly, $\nu_{sum}$ should have the same value (not only the same asymptotic
behavior) across all AON versions in order to obtain consistent results. The average probability $\bar{p}_{crit}$ scales, not surprisingly,
linearly with $\delta t$. For QuadSamp, $\bar{p}_{crit}$ is inversely proportional to $N_{SIP}$ and an increase of $N_{SIP}$ decreases the occurence of
multiple collections and limiter events. In the LinSamp case, $\bar{p}_{crit}$ is independent of $N_{SIP}$ (as already pointed out by Shima
et al., 2009, end of their section 5.1.3) implying that an increase of $N_{SIP}$ does not decrease the number of multiple collections
and limiter events. Nevertheless, an $N_{SIP}$-increase is also beneficial in LinSamp as it increases the number of trials and reduces
the variance of the results.
## 3.4  Full column model simulations
The box model emulation simulations presented in Sec. 3.2 used an academic and irrealistic setup, not yet exploiting the
capabilities of a column model framework. The following two subsections treat realistic setups.
### 3.4.1  Half domain setup
We initialise droplets in the upper half of a $4\,\text{km}$ column. In each GB the mean radius of the DSD is fixed at the default value
$r_0 = 9.3\,\mu\text{m}$. $LWC$ (and with it $DNC$) decreases linearly from $3\,\text{g/m}^3$ at the model top to zero at $z = 2\,\text{km}$. At the model
top, a constant influx of a DSD with $LWC = 3\,\text{g/m}^3$ is prescribed which guarantees a smooth profile over time. Otherwise, a
discontinuity would occur at the top-most GB which may raise problems in the bin model.
The further settings are $nz = 400$, $\Delta z = 10\,\text{m}$, $\Delta t = 10\,\text{s}$, $nr_{inst} = 20$, $\kappa = 40$. Figure 15 shows the temporal evolution of
the mean diameter and the moments $\lambda_0, \lambda_1$ and $\lambda_2$. Due to the influx condition, the total mass increases during the first 10 min-
utes, barely visible in the third panel. During this period, however, collection is already efficiently reducing the droplet number.
This is accompanied by an increase of the mean diameter and radar reflectivity. Soon after, the first droplets reach the surface,



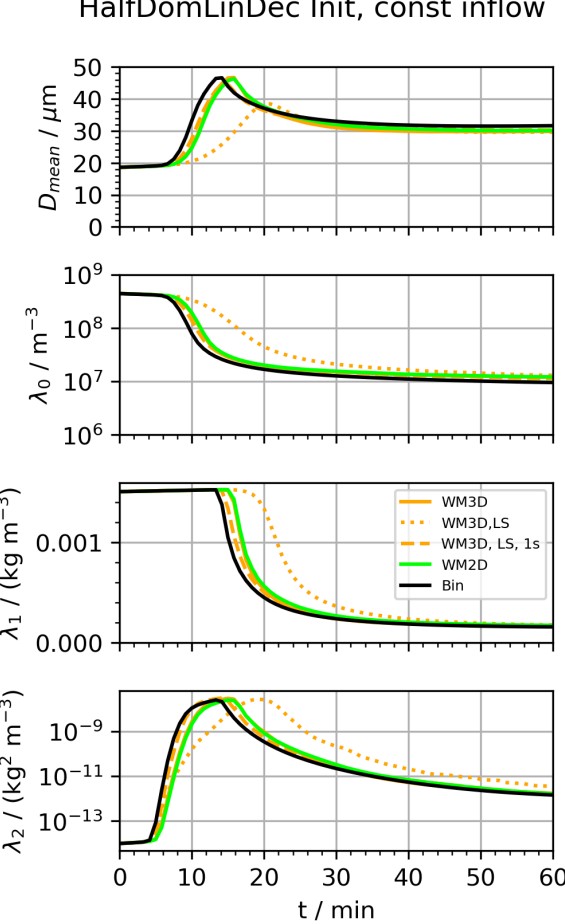

**Figure 15. HalfDomLinDec setup:** Temporal evolution of $D_{mean}$ and column-averaged moments $\lambda_0, \lambda_1$ and $\lambda_2$ for various model versions (see inserted legend; "LS" is short for linear sampling). .

the mass declines rapidly, and the whole column is more or less washed out after 30 minutes. We find an excellent agreement
among the three model versions BIN1D, AON-WM3D and AON-WM2D. Using LinSamp in AON-WM3D, agreement with
the other models is reached only if the time step is reduced (here from $\Delta t = 10\,\mathrm{s}$ to $1\,\mathrm{s}$).
Figure 16 shows vertical profiles of $DNC, LWC, Z$ and $N_{SIP,GB}$ for times $t = 0, 10\,\mathrm{min}, 20\,\mathrm{min}, 30\,\mathrm{min}$ and $60\,\mathrm{min}$. In the
upper half, droplet number is roughly homogeneously distributed and decreases over time. In the lower half, droplet number
concentrations are several orders of magnitude smaller than in the upper half and increase over time. The profile of the radar
reflectivity shows the highest values after 10 minutes with a pronounced peak in the middle of the domain. Soon after, the
$Z$-profiles become smooth and increase monotonically towards the surface. The sedimentation flux also increases towards the
surface and hence $\lambda_2$-values decrease over time.



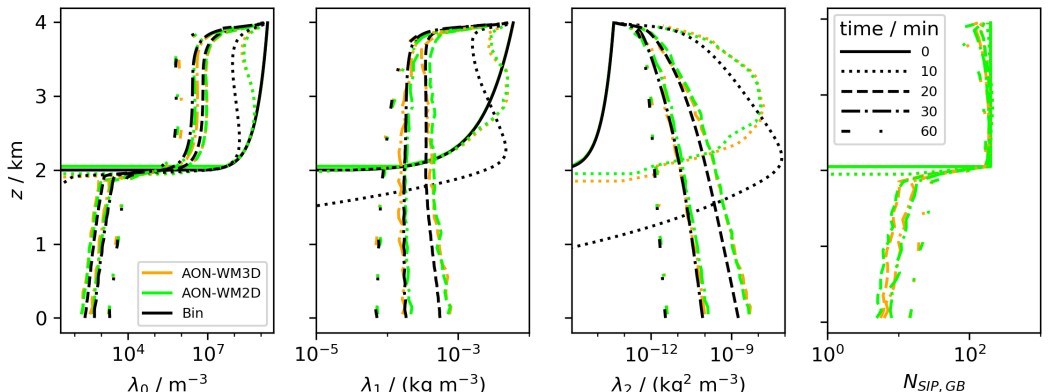

**Figure 16. HalfDomLinDec setup:** Vertical profiles of moments $\lambda_0, \lambda_1, \lambda_2$ and $N_{SIP,GB}$ for various model versions (AON-WM3D, AON-WM2D, Bin; see color legend in left-most panel) and times ($0, 10, 20, 30, 60$ min; see linestyle legend in right-most panel).

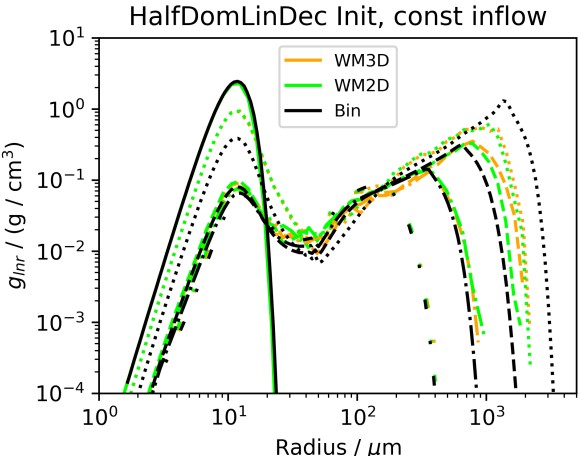

**Figure 17. HalfDomLinDec setup:** Size distribution $g_{lnr}$ for various model versions and times as in Fig. 16 (see legends there).

In the upper half, $N_{SIP,GB}$ is fairly constant over altitude and time with around 200 SIPs. As the $LWC$ is initially highest
at the model top, collections are most frequent there. Most likely, SIPs from that layer turn into collector SIPs and fall through
the total column. Consistently, $N_{SIP,GB}$ decreases over time close to the model top. Yet overall, only a small fraction of the
SIPs becomes rain drops eventually (see e.g. Fig. 4 in U2017) and hence the SIP number is substantially smaller in the lower
half. There, each GB is populated roughly by 10 SIPs. Despite this rather small value, convergence in $DNC$ and $Z$ seems to
be ubiquitous.
Figure 17 depicts column-averaged DSDs for various points in time. The precipitation mode develops rapidly, and 2 to 3 mm-
sized drops are produced within 10 minutes. Those drops soon reach the surface and remove a significant amount of liquid





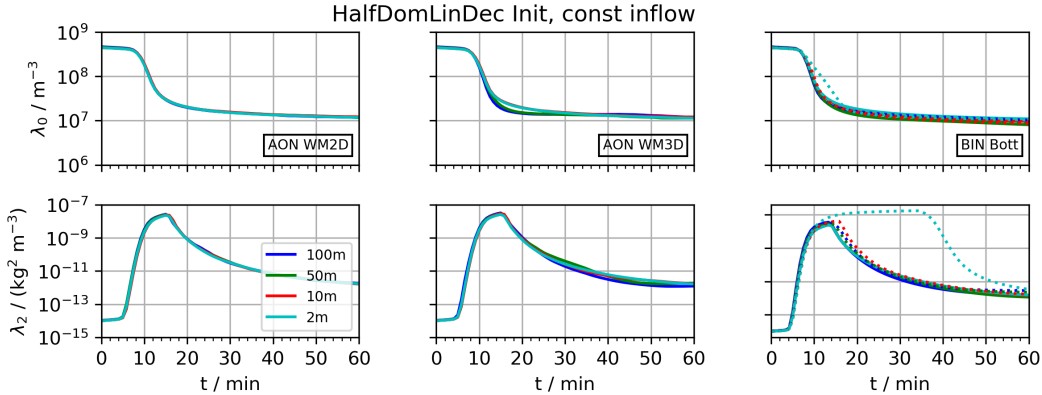

**Figure 18. HalfDomLinDec setup:** Temporal evolution of column-averaged moments $\lambda_0$ and $\lambda_2$ for various model versions (AON WM2D, left; AON WM3D, middle; Bin, right). Each panel shows a variation of the vertical resolution $\Delta z$ (see legend). In LCM simulations, SIP numbers for $\Delta z = 100\,\mathrm{m}$ and $50\,\mathrm{m}$-simulations are increased to the level of the $\Delta z = 10\,\mathrm{m}$-simulation. The right column shows MPDATA (solid) and US1 (dotted) results.

water from the column. Due to this wash-out effect, the rain drops cannot grow that large any longer and the precipitation
mode peaks at smaller sizes at later times. Overall, the agreement between the three model versions is remarkable given the
completely different numerics of the Eulerian and Lagrangian approach.

4       Next, the vertical resolution $\Delta z$ is varied in all three model versions. Even though this sounds like a banal sensitivity study,

the effect of a $\Delta z$-variation has different implications in the various model variants and the differences are rather subtle. First,
$\Delta z$ affects the number of GBs $nz$ and with it the total SIP number $N_{SIP,tot}$ (as $N_{SIP,GB}$ is unchanged with the standard
SIP init technique). To eliminate this unwanted numerical side effect in LCM1D, we increase $N_{SIP,GB}$ proportionally to $\Delta z$
(analogous to the $\Delta V$-sensitivity tests in section 3.2). Second, the advection by sedimentation changes in BIN as the CFL
number changes and the subcycling has to be adapted. In LCM1D, the SIP transport by sedimentation is independent of the
assumed grid and clearly unaffected by a $\Delta z$-variation. Third, there is a physical effect as $\Delta z$ determines the layer depth of the
well-mixed volume (effective only in AON-WM3D and BIN).
It follows that the results of the AON WM2D version should be independent of $\Delta z$. Moreover, the AON-WM3D variant can
be used to determine if the size (more specifically the depth) of the well-mixed volume is a crucial parameter. In bin models in
general, the latter effect could not easily be singled out as sedimentation numerics also change with $\Delta z$.
Figure 18 depicts the evolution of $\lambda_0$ and $\lambda_2$ for $\Delta z$ ranging from $2\,\mathrm{m}$ to $100\,\mathrm{m}$. As expected, the AON WM2D simulations
are not at all affected by $\Delta z$ (left column). The middle column shows the AON-WM3D simulations. The $\Delta z = 10\,\mathrm{m}$ simulation
uses $N_{SIP,GB} = 200$ and the $\Delta z = 100\,\mathrm{m}$-simulation $N_{SIP,GB} = 2000$. Hence, a factor 100 more SIP combinations are tested
for possible collections in the latter case. Nevertheless, the results are basically identical, implying that the depth of the well-
mixed volume has a negligible impact on the extent of collections in the present example. The right column shows the BIN
results which are again basically identical, using the MPDATA scheme (solid) and the 1st order upwind scheme (dotted). The



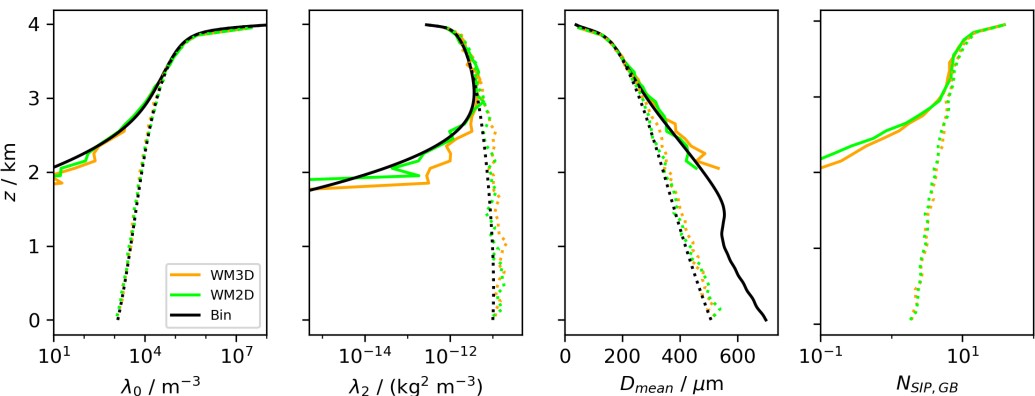

**Figure 19. EmptyDom setup:** Vertical profiles of moments $\lambda_0, \lambda_2, D_{mean}$ and $N_{SIP,GB}$ for various model versions (see legend). Depicted are the times $t = 30$ and $60$ minutes (solid, dotted).

slight deviations in $\lambda_0$ may be due to the fact, that in a bin model the vertical redistribution by sedimentation is also affected
by $\Delta z$. Due to stability issues, the time step (for collection) had to be reduced from $\Delta t = 10\,\mathrm{s}$ to $1\,\mathrm{s}$ for US1. Then, reasonable
results are achieved for $\Delta z \geq 10\,\mathrm{m}$. For the highest resolution $\Delta z = 2\,\mathrm{m}$, however, numerical instabilities are still present (see
outlier curve). This is a clear indication for the superiority of MPDATA in BIN.
**3.4.2   Empty domain setup**
In this section, the $4\,\mathrm{km}$ deep column is initially devoid of droplets and a time-constant influx of a DSD with $r_0 = 16.9\,\mu\mathrm{m}$ and
$LWC = 6\,\mathrm{g/m^3}$ is prescribed. As in the box model emulation setup, the according DNC is $297\,\mathrm{cm^{-3}}$.
Over time the column fills with droplets, a distinct size sorting is established and DSDs at a specific altitude are expected to
be rather narrow. Hence, choosing a too coarse vertical resolution may result in overestimating collections as the droplets are
not supposed to be well-mixed within such deep GBs. In such a case, the AON WM2D variant has a conceptional advantage as
it does not assume well-mixedness in the vertical direction. The chosen setup specifically aims at demonstrating the possible
improvement of this. Again, the further parameter settings are $nz = 400$, $\Delta z = 10\,\mathrm{m}$, $\Delta t = 10\,\mathrm{s}$, $nr_{inst} = 20$, $\kappa = 40$.
Figure 19 shows vertical profiles at $t = 30$ and $60$ minutes. After 30 minutes the cloud roughly covers the top half of the
column. Below $z = 2\,\mathrm{km}$, fewer than 0.1 SIPs are present in each GB of LCM1D. This implies that only in 1 or 2 out of
the 20 realisations SIPs grow sufficiently large to fall that far. This also explains the jagged $\lambda_2$-profiles in the lower part.
Below a certain altitude, no SIPs are present at all and hence no mean droplet diameter could be diagnosed. BIN produces
non-zero mass and number all the way down to the bottom and allows computing a smooth $D_{mean}$-profile. As the predicted
droplet masses become vanishingly small, the derived $D_{mean}$-values in the lower part are, however, meaningless. Anyhow, this
small discrepancy between BIN and LCM1D is a transient phenomenon. Once the cloud is fully developed, the profiles match
perfectly (see dotted curve for $t = 60\,\mathrm{min}$). Remarkable is the fact that on average well below 10 SIPs populate GBs in the



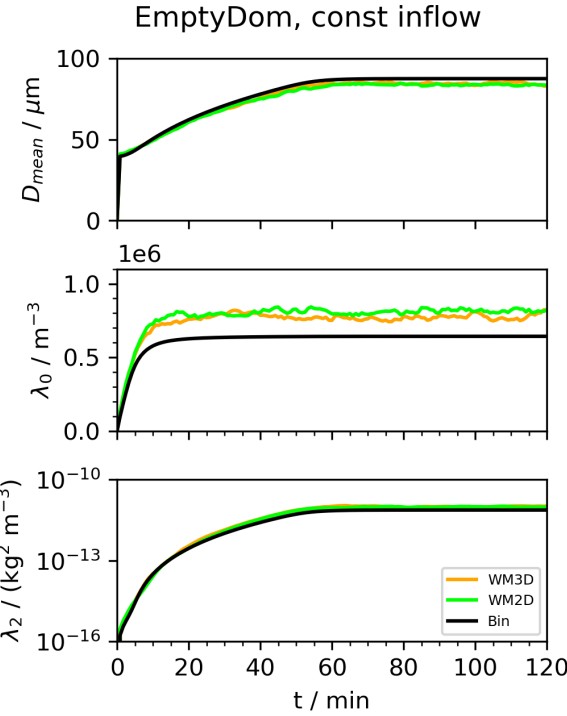

**Figure 20. EmptyDom setup:** Temporal evolution of $D_{mean}$ and column-averaged moments $\lambda_0$ and $\lambda_2$ for various model versions (see legend).

lower domain half. Nevertheless, the LCM1D results seem to be converged. SIPs at those altitudes are large ($D_{mean} > 400\,\mu$m)
and fall fast, which fosters a strong SIP exchange across GBs and is beneficial to convergence (see section 3.2).
Figure 20 shows the temporal evolution of the mean diameter, column-averaged $DNC$ and $Z$. Within the first 10 minutes,
DNC increases quickly. Soon after, collection becomes effective and DNC reaches a quasi steady state. The radar reflectivity
increases within the first 60 minutes and then also reaches a quasi steady state. The only discrepancy between the various
models are slightly larger DNC-values with LCM1D. The reason for this is elucidated next.
Fig. 21 shows the $\Delta z$-dependence of the DNC and Z-evolution in the different models. For $\Delta z = 50$ and $100\,$m, the SIP
numbers in LCM1D have been upscaled to maintain $N_{SIP,tot}$-values comparable to the $\Delta z = 10\,$m-simulation (as already done
in the HalfDom-setup). The $Z$-evolution (second row) is found to be basically independent of $\Delta z$ in all three models. For the
DNC-evolution, we find also no $\Delta z$-dependence in the WM2D-model as intended. However, in WM3D and BIN model, DNC
levels off at different values depending on $\Delta z$. This latter behavior is most likely caused by an interaction of the unresolved
size sorting and the hence larger range of potential collection partners in AON-WM3D and BIN. Apparently, this results in
changes in the rate with which the smallest droplets are collected by larger droplets, as indicated by the substantial effect of
this process on DNC but not on Z.

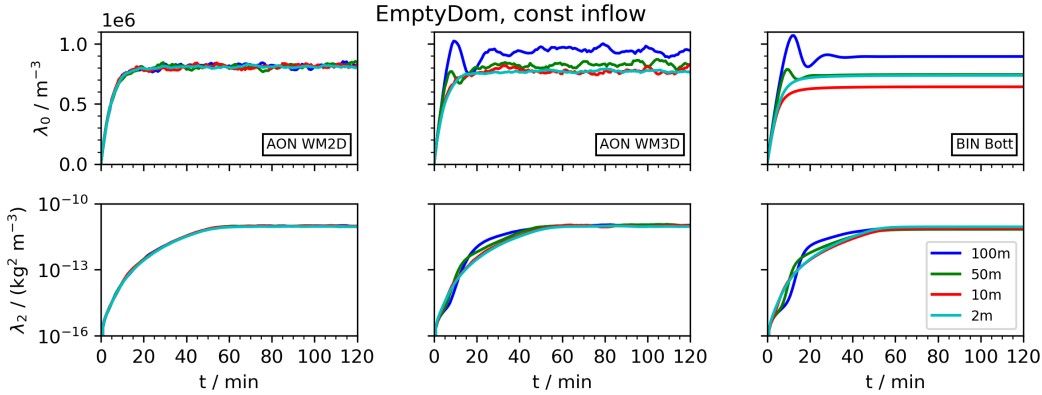

**Figure 21. EmptyDom setup:** Temporal evolution of column-averaged moments $\lambda_0$ and $\lambda_2$ for various model versions (AON-WM2D, left; AON-WM3D, middle; Bin, right). Each panel shows a variation of the vertical resolution $\Delta z$ (see legend). In LCM simulations, SIP numbers for $\Delta z = 100\,\mathrm{m}$ and $50\,\mathrm{m}$-simulations are increased to the level of the $\Delta z = 10\,\mathrm{m}$-simulation.

For $\Delta z = 100\,\mathrm{m}$ and $\Delta z = 10\,\mathrm{m}$, Figure 22 shows the DNC-evolution of the WM3D-model with different parameter set-
tings. The green curves shows the default case from before, where the $\Delta z = 100\,\mathrm{m}$-simulation uses a "10x" higher $N_{SIP,GB}$-
value. We used LinSamp instead of QuadSamp (red), further decreased the time step from $\Delta = 10\,\mathrm{s}$ to $1\,\mathrm{s}$ or used for both
resolutions the same $N_{SIP,GB}$-value (which reduces $N_{SIP,tot}$ of the $\Delta z = 100\,\mathrm{m}$-simulation by a factor of ten). In all cases,
the $\Delta z$-dependence appears consistently in all parameter settings.
This undesired $\Delta z$-dependence in BIN and WM3D seems to showcase the superiority of the AON-WM2D implementation.
However, the $\Delta z$-dependence does not affect higher moments of the DSD, e.g., Z (Figs. 20 and 21) or the accumulated size
distribution of all droplets that crossed the lower boundary (Fig. 23). Accordingly, precipitation-related quantities seem to be
unaffected by changes in the vertical grid spacing. On the other hand, most of the $\Delta z$-effect can be attributed to changes in the
DNC within the top most $100 - 200\,\mathrm{m}$ of the column (Fig. 19), which might affect the radiative properties of the considered
cloud. Anyhow, we cannot definitely answer the question, whether using the AON-WM2D approach has any practical benefits
over the classical 3D well-mixed approaches based on the presented results. Further research is required.

## 4  Summary and conclusions

Collection, i.e., the coalescence, accretion, and aggregation of hydrometeors, is an important process for the development
of precipitation in liquid-, mixed-, and ice-phase clouds, respectively. Moreover, aggregation leads to irregular ice crystal
shapes affecting the cloud radiative properties. The correct representation of these processes in cloud microphysical models
is, therefore, of utmost importance. In this study, we investigated and validated the representation of collection in LCMs, a
relatively new approach that uses simulation particles, so-called SIPs or superdroplets, to represent cloud microphysics.



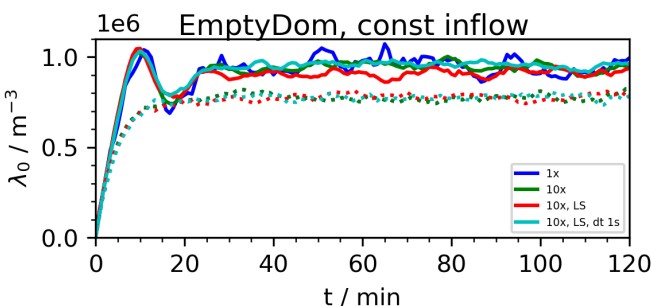

**Figure 22. EmptyDom setup:** Temporal evolution of column-averaged moments $\lambda_0$ and $\lambda_2$ for the AON-WM3D model. Results for various parameter settings (see legend) are depicted for $\Delta z = 100\,\mathrm{m}$ (solid) and $\Delta z = 10\,\mathrm{m}$ (dotted).

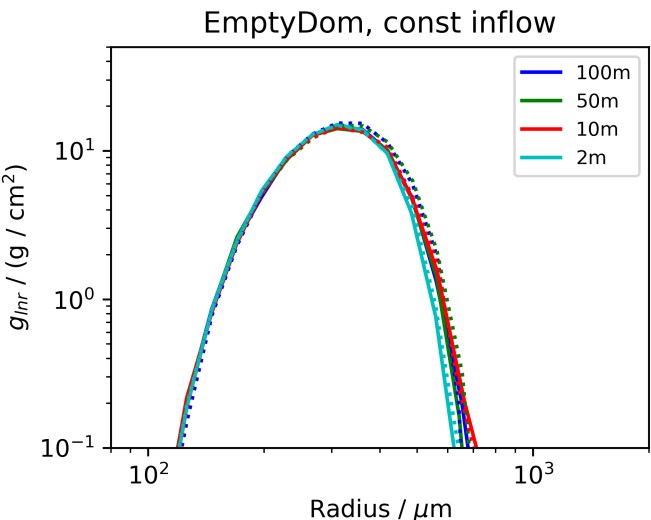

**Figure 23. EmptyDom setup:** Size distribution of all droplets that crossed the lower boundary. AON-WM3D (dotted) and AON-WM2D (solid) results for various vertical resolutions $\Delta z$ are displayed (see inserted legend for the color coding).

This study is a continuation of U2017, in which we analyzed various representations of collection in LCMs using zero-
dimensional box model simulations. Here, this analysis is extended to one-dimensional column simulations that allow consid-
ering the effects of sedimentation explicitly. This study focuses on the AON collection algorithm (Shima et al., 2009; Sölch
and Kärcher, 2010) that outperformed other collection approaches, as assessed in our previous study (U2017). Two variants of
AON are applied that differ in the assumed distribution of droplets represented by a SIP: In WM3D, the droplets are assumed
to be well-mixed within a three-dimensional volume (which is typically identical to the GB of the dynamical model coupled
to the LCM). In WM2D, the height coordinate of each SIP is used explicitly, and the droplets represented by a SIP are as-
sumed to be well-mixed only within a two-dimensional, horizontal plane. Accordingly, collections are only considered if a SIP





overtakes another one during a timestep. Furthermore, two variants of AON-WM3D are tested that differ in the number of SIP
combinations that need to be tested during collection. In its simplest form, AON-WM3D depends quadratically on the number
of SIPs since every SIP may interact with any other SIP inside a GB (QuadSamp). Additionally, Shima et al. (2009) introduced
an approach that depends only linearly on the number of SIPs by appropriately scaling collection probabilities (LinSamp).
All results are compared to established Eulerian bin model results (Bott, 1998; Wang et al., 2007). Accordingly, the capability
of Lagrangian and Eulerian approaches to advect a droplet ensemble due to sedimentation is tested first — neglecting the
influence of collection. Since numerical diffusion is inherent to any Eulerian advection problem, i.e., also sedimentation, its
impact might impede any conclusions drawn from the collection simulations. However, by using an appropriate advection
scheme (MPDATA, Smolarkiewicz, 1984), numerical diffusion can be reduced to an acceptable degree in the sense that the
present simulations focus on the differences driven by collection numerics.
To bridge the gap to U2017, the behavior of box model simulations is emulated in the column model. This is done by
initialising each GB of the column with the same droplet size distribution and applying cyclic boundary conditions at the
surface and the top. By using this framework, we were able to show that sedimentation increases the model convergence
rate significantly compared to box model simulations without sedimentation, i.e., significantly fewer SIPs are required in the
column model. The reason for this behavior is that the largest and hence fastest falling droplets are no longer confined to the
same GB and to the same potential collection partners, hence increasing the ensemble of potential collection partners. A similar
observation has been made by Schwenkel et al. (2018), who used randomized motions between individual GBs. Overall, these
results indicate that a simulation with only 24 SIPs per GB can yield reasonable results if (i) these SIPs are able to move
between GBs and (ii) the SIP weighting factors are ideally chosen in the beginning by using an approriate SIP initialisation
technique.
A generally good agreement of the LCM results with the bin reference has been found for all AON variants. However,
they reveal distinct differences in their numerical and computational requirements. LinSamp demands a shorter timestep than
QuadSamp as a result of the upscaled collection probabilities to avoid SIPs with a zero (or even negative) weighting factor.
And indeed, fully coupled LCM applications with AON and LinSamp are reported to require a relatively short timestep to
reach convergence (e.g., Dziekan et al., 2019). Accordingly, these strong restrictions on the timestep might cancel out the
computational benefit gained by the reduced number of SIP combinations that need to be tested in LinSamp. This indicates
that the simpler QuadSamp might be a valuable alternative to LinSamp as long as the number of SIPs is not prohibitively high.
We further compared the computational requirements for the WM2D and WM3D implementations of AON. We found that
WM2D requires to check for overtakes in the entire column, not only in the GB in which the SIP is located, as is the case for
WM3D. However, this seeming disadvantage is turned into an advantage, since only a minority of SIPs overtakes other SIPs.
Accordingly, the overall number of calculations necessary for the application of WM2D is reduced compared to WM3D. The
physical reason for this effect is the typical bimodal structure of droplet spectra, which consist of only a few large droplets that
sediment and collect other droplets efficiently, while the remaining droplets are usually too small to sediment and collect other
droplets.





Finally, we applied these approaches to two more realistic column cases. While both cases use a prescribed inflow of droplets from the top, the first case is initialised with a linearly increasing liquid water content, and the second case is completely devoid of any initial droplets. Overall, the agreement of AON-WM3D, AON-WM2D, and the bin references is remarkable. Only in the second case, which is designed to be heavily prone to size-sorting, a dependence on the vertical grid spacing is detectable for WM3D and the bin reference, which both assume the droplets to be well-mixed within a GB, while the WM2D results are found to be completely independent of the vertical grid spacing.

All in all, this study has shown that the representation of collection in LCMs using AON with WM3D and WM2D reproduces established Eulerian bin results successfully. This ability, of course, depends foremost on the number of SIPs and the applied timestep as already indicated in previous zero-dimensional box model studies. Compared to these zero-dimensional studies, the application of an LCM in a column decreases the required number of SIPs significantly. The consequently lower computational costs raise hopes to use LCMs more frequently in large-scale, multidimensional models in the future.

*Code and data availability.* The source code of the Lagrangian column model is hosted on GitHub (https://github.com/SimonUnterstrasser/ColumnModel). The (frozen) code version used to produce the simulation data of this study can be obtained from Zenodo (DOI: 10.5281/zenodo.3547539). The data of the BIN and AON simulations together with all plot scripts, which are necessary necessary to reproduce the figures of this study, are released in a second Zenodo data set (DOI: 10.5281/zenodo.3547341)

*Author contributions.* S. Unterstrasser designed the study, programmed the Lagrangian column model, carried out the simulations, wrote most parts of the manuscript. F. Hoffmann discussed the results with the first author and wrote the introduction and conclusions. A first code version and preliminary results were obtained during the Master's thesis of M. Lerch.

*Competing interests.* The authors declare that they have no conflict of interest.

*Acknowledgements.* We thank L.P. Wang and A. Bott for providing box model versions of their collection algorithms. The first author thanks Jan Bohrer (Tropos Leipzig) for carefully examining the AON code and spotting the bug mentionned in section 3.2.1. Moreover, we appreciate comments on the manuscript by K. Gierens. This research was performed while Fabian Hoffmann held a Visiting Fellowship of the Cooperative Institute for Research in Environmental Sciences (CIRES) at the University of Colorado Boulder and the NOAA Earth System Research Laboratory.





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
