# Peer review of "Collisional growth in a particle-based cloud microphysical model: Insights from column model simulations using LCM1D (v1.0)"

_Geoscientific Model Development, 2019_

## Referee Comment (RC1) · Anonymous Referee #1 · 9 Mar 2020

Authors of this paper test the behaviour of a numerical algorithm representing collisions between cloud and rain drops in Lagrangian microphysics schemes. They describe 3 versions of the algorithm (one of the versions is new) and test the convergence of the 3 versions in 3 different settings.

The presented work is based on a similar suite of tests carried out by the same group of authors and published in GMD in 2017 (https://www.geosci-model-dev.net/10/1521/2017/gmd-10-1521-2017.html). In 2017 the tests were done in a 0-dimensional box setup. Now the tests are extended to a 1-dimensional column.

I think that the presented tests are useful and the topic is interesting to the GMD com-

munity. However, the overall presentation of the convergence of the algorithm should be improved before publication.

I couldn't find what is the criterion for reaching convergence in the tests. Despite being very thorough in testing different parameters and cases, the authors then use vague terms like "equally bad", "tend to approach the reference" or "seem to converge". The only way in which the authors show convergence is by plotting many lines on top of each other. This is not satisfying and results in a paper that is overflowing with figures that look the same. I suggest introducing a definition and quantitative measure of convergence. This would allow to change some of the repetitive concentration plots into plots showing convergence as a function of a tested parameter.

I think that the paper should be more concise. The authors carried out a lot of tests but summarising some of them would improve the paper. For example Fig 10 shows results for slightly different initial condition and Fig 12 for a different kernel. Neither of these figures show anything new about the collision algorithm behaviour. Similarly, the authors study the a bin collision algorithm with upstream advection scheme and observe that the results are "slightly smeared out". The upstream advection scheme is known to be very diffusive and there were many papers published on that. This detracts from the main theme of the paper and the interesting parts of the study.

Additional comments:

Table 1: German language in the caption

Page 3, line 13-14: "Moreover, we will use the term cloud droplets interchangeably with ice crystals to increase clarity in writing." - I don't think that this increases the clarity. The paper nowhere actually discusses issues related to ice crystals. I think that keeping the language focused on cloud and rain droplets is sufficient.

Figure 1: in SIP (Simulation particle) p should also be capitalised

Page 7 line 15: Smoluchowski equation

Page 8: Maybe it would make more sense to first describe the collision algorithms and then talk about the column model setup?

Figure 2,3 and later in the text: The abbreviations like AON, WM3D, etc were already defined and should not be defined again.

Page 9 line 5: I don't think it's legally allowed and generally acceptable to copy verbatim paragraphs from different papers (?). I would suggest to just refer to the relevant paragraph or to paraphrase.

Algorithm 1 and 2 caption: The style convention of the code block should be repeated in its caption.

Algorithm 2, line 16: iff

Page 14 line 15: "For more sophisticated kernels, including, e.g., turbulence enhancement, the present approach may not be adopted easily as the driving mechanism for collisions to occur in the current model is differential sedimentation (...)." - This is very important for real applications of WM2D algorithm. Could you expand on this? How would you implement the WM2D ideas in a full 3D LES simulation focusing on turbulence effects on precipitation formation?

Paragraph 2.5: Please provide a table which summarises the combinations of different algorithm options and the labels used to distinguish them.

Paragraph 2.5: What is the added benefit of comparing the BIN Bott and Wang collection algorithms in a paper about Lagrangian collection algorithms?

Page 19 line 8: "We found that convergence is usually more easily reached for higher moments than for lambda_0" – Why is that? Would different method of initialising the SIPs make a difference?

Figure 7 and others: Could you show results for t> 30min only? The first half of all those plots shows nothing.

[Figure]

Figure 7 and others: Why are the Lagrangian schemes always predicting higher concentrations than the BIN scheme?

Figure 14: This figure is very interesting! It's staggering to see how very few of the tested combinations of SIPs lead to any collisions. A strong argument in favour of WM2D approach.

Figure 16: The figure is not clear, especially the yellow dash-dotted lines are hard to see.

Page 34 lines 15-19: If the results are "identical" and "basically identical" then they don't have to be shown again.

Page 35 line 4: This is not a new finding and is out of scope of the paper.

Figure 21: It's very hard to distinguish between the two blue lines.

Page 40 line 14: necessary is repeated

Code availability: It would help to provide a Docker or a Singularity image in which the column model scripts could be run. It would eliminate the need to change any file paths or to install compilers and packages with specific versions. This would be helpful, especially because the provided code is not just pure Python code. I know that it is not a policy of GMD, so it's just a suggestion.

In my case I tried compiling the code with gcc 8.3.0 and gcc/9.2.0 on a CentOS system with Python 3.7.0 and got an error:

AON_Alg.fpp:90:0: error: operator '*' has no right operand #if (KERNEL_INTPOL <= 1) /* logarithmic mass bin*/

I suspect it is some preprocessor issue but I didn't debug further. I then tried compiling with gcc/9.2.0 on an OSX system with Python 3.7.6 and got an error:

sed: illegal option – -

[Figure]

I guess it's an issue with OSX default sed, but I didn't debug further.

---

## Referee Comment (RC2) · Sylwester Arabas (Referee) · 1 Apr 2020

**Review of the GMD-2019-343 manuscript**

Hereby I provide my comments to the manuscript entitled "Collection/Aggregation in a Lagrangian cloud microphysical model: Insights from column model applications using LCM12 (v0.9)"

Overall, I consider the submitted text a valuable contribution to the literature on particle-based cloud modelling as it provides detailed formulation of test cases that are essential in the development of new implementations of relevant algorithms. It thus clearly matches the journal scope.

Below, I list my major, minor and technical comments to the manuscript. I include also a comment to the enclosed software.

**Major points**

**Unacknowledged performance trade-offs**

Some performance trade-offs pertaining to the choice among linear and quadratic sampling are detailed. It is however not pointed out that linear sampling precludes parallelisation of the collision computations (within a gridbox/column) due to introduced data dependency (page 11, line 22-23). This is particularly worth underlining, as the availability of shared-memory parallelisation with multi-core CPUs or GPUs allows for significant speed up (i.e., almost by the factor equal to the number of threads). In fact, all but the pair-shuffling and random number generation steps in the AON coalescence algorithm with linear sampling are embarrassingly parallel. Given the above, I find it at least misleading to say, without mentioning the precluded parallelisability, that:

- "simulations with linear sampling ... converges slower ... compared to quadratic" (p1/l14-15)

- "benefit of the reduced computational cost may be outweighed by the stronger requirements on $\Delta t$" (p24/l2)

- "restrictions on the timestep might cancel out the computational benefit gained by the reduced number of SIP combinations" (p39/l25-26)

Similarly, I doubt the statement on page 12, lines 2-3 (on performance superiority over integer-preserving implementation) holds true in parallel context, where random numbers can be generated concurrently in large batches.

Finally, it is worth commenting on the parallelisability consequences of the requirement to perform collisions column-wise in the WM2D scheme.

**Subrid-dynamics and WM2D**

While it is acknowledged on page 14 (lines 15-17) that the WM2D scheme is somewhat incompatible with "sophisticated kernels", I highly recommend to extend the discussion also to the aspects of subgrid-scale dynamics representation in particle-based models – e.g., referring to the already cited work of Hoffmann et al. (2019). In short, in my understanding, the "information content" of SIP positions in particle-in-cell-type models is somewhat overestimated here. In particular, the prevalent Large-Eddy-Simulation context should be addressed. Candidate location: page 14 (lines 11-14)?

**Paper length**

The article length is, in my opinion, impeding appreciation of its content. I include some detailed suggestions on what could be omitted from the text in Technical/editorial remarks below. Besides that, I consider it a malpractice to introduce an almost-page-long quote from an earlier study of the authors. I see also little benefit in repeating Figure. 3 here – please just refer to the relevant parts of the 2017 GMD paper which is readily available for all readers.

I also suggest adding a table of contents (as done recently in GMD in Shima et al. [3]).

**Minor points**

**The title**

First, why not to avoid a "slash" in the title, and use "Collisional growth" instead of "Collection/Aggregation". Second, I oppose to calling presented simulations "applications", suggest "simulations"?

Finally, I generally suggest to label the discussed microphysics modelling methods as "probabilistic particle-based" rather than "Lagrangian". First, "Lagrangian" is a much more broader term (consider e.g. the Lagrangian cloud models described in [2, 1]), and thus potentially misleading for readers from outside our niche. Second, the discussed model is not fully Lagrangian as it relies on Eulerian dynamical core.

I am aware that the present title is a reference to the U2017 paper, but perhaps the above arguments outweigh it?

On a related note, there is not a single mention of the "Monte-Carlo" keyword in the paper, please do cater to a wider community and use such keywords to give a good context for the readers.

**The limiter**

The ad-hoc definition of the "limiter" (p15/l6) calls at least for a reference to the min() in the SDM paper's $\tilde{\gamma}_\alpha := \min(\gamma_\alpha, [\xi_{j\alpha}, \xi_{k\alpha}])$ expression (Shima et al. 2009, step (5) in the left column), if not for reformulating the "limiter" in a more robust manner.

**Integer vs. real-valued weighting factors**

I do not find enough grounds in the text for the statements on the superiority of real-valued vs. integer-valued weighting factors. Besides the above commented issue of parallel random number generation/multiple collisions, the statements on page 11, lines 30-31 seem to overlook the concept of spectrum estimation, see e.g. the third paragraph in section 5.1.4 in (Shima et al., 2009) (also, worth mentioning when discussing eq. 14).

**CFL condition for sedimentation**

I would argue that we should assume the CFL condition for particle sedimentation as well – while it does not cause the numerics to fail as in Eulerian component, it is intuitively not fulfilling the assumptions (as acknowledged on p4/l12). Relevant statements: p8/l11, p12/l26, p21/l15, p34/l10.

**Courant number values in section 3.1**

Upwind and MPDATA convergence is dependent on the Courant number – please indicate which was used in section 3.1. Perhaps worth checking the behaviour for a set of Courant numbers.

**Correlations**

Please be more specific as to the mathematical meaning of "correlations" mentioned 12 times in the context of collisions but never defined.

**Classical/regular nomenclature**

References to "classical/regular" "implementations/approaches/cases/versions/AON" (p8/l15, p8/l16, p8/l22, p8/l24, p9/l4, p11/l12, p12/l8, p12/l19, p12/l29, p12/l31, p14/l7, p14/l8, p14/l30, p15/l4, p17/l2, p19/l1, p25/l5, p25/l21, p29/l1, p30/l2, p37/l12) are not understandable, especially given that the authors introduce their own nomenclature for numerous notions named differently in literature. I am aware that there are some definitions of "regular" in the text, but it is an over 40-page long paper. Please come up with more precise and less subjective statements.

**Technical/editorial remarks**

**Figures**

I urge the authors to replace raster images in figures 1 and 4-23 by its vector-format equivalents (i.e., `plt.savefig(format='png', ...)` $\rightsquigarrow$ `plt.savefig(format='pdf', ...)`). I suggest using $cm^{-3}$ as the unit for $\lambda_0$ on the plots.

**Text**

**p1/l2** the "high number" is equally (or even more) applicable to bin models, please rephrase and indicate with respect to which benchmark the value is high

**p1/l4** is the word "explicit" needed (suggest avoiding if the opposite "implicit" is not clear)

**p1/l8** ditto

**p1/l12** "accuracy" $\rightsquigarrow$ "resolution"

**p2/l28** please underline that it is you who introduce the AON term

**p2/l22** why not in chronological order? (see also background works listed in Shima et al. [3] and dating back to 2004)

**p2/l26** "abbreviated as ... in the following" $\rightsquigarrow$ "hereinafter abbreviated ..."?

**p3/Table 1** please remove non-English caption, and consider removing the table – given the multitude of symbols used in the text, it seems anecdotal to list 8 abbreviations in a table (moreover, the following are not listed: DNC, MPDATA, CFL, WM2D, WM3D, MC, BIN, US1, noSedi, LCM0D, LCM1D, LS, ...)

**p3/l3** suggest removing/rephrasing "relatively young modelling approach" – particle-in-cell method is 50-year old; same for Monte-Carlo for collision

**p3/l4** "aws" ⇝ "was"

**p3/l12** "coalescence, aggregation, or accretion" – mention "riming", "self-collection", "wash-out" keywords as well?

**p3/l13-14** actually, ice crystals are mentioned only one after this statement – suggest removing

**p3/l22** "(sometimes pedantically)" sounds negative ... suggest not being pedantic (see below) and removing the statement

**p4/l3** suggest using subscripts for $L_z$ and $n_z$

**p4/l6** no need to make it a separate numbered equation?

**p4/l8** no need to define volume of the sphere - just mention in the text

**p4/l12** $K(m_i, m_j)$ and $K(r_i, r_j)$ would likely be better named differently

**p4/l15** "radius-dependent" ⇝ "sie-dependent"

**p4/l17** "latter" ⇝ "last" (there are three assumptions listed before)

**p4/l18** there is a bogus character before 500 $\mu$m - garbage displayed in the pdf viewer I'm using

**p5/l1** "collection" ⇝ "collisional growth"?

**p5/l9** "latter" ⇝ "last" (there are three assumptions listed before)

**p5/l12** no need to define factorial

**p5/l13** skip reference to Berry 1967, surely "mass density function with respect to the logarithm of radius" is enough

**p5/l22** first sentence is a repetition from the Introduction

**p5/l24-25** suggest rephrasing around "terms low and high"

**p6/l14** random interval should be [...) and not [...] (as per numpy.random docs)

**p6/l14** "some threshold" – please be specific (also, worth mentioning the alternative formulation with integers)

**p7/l3** "around" – please be specific

**p7/l6** random number interval: [...)

**p7/l6-7** unneeded sentence (Furthermore...)?

**p7/l15** rephrase "to solve the Smoluchowski."

**p7/l18** use partial derivatives

**p7/l21** constant-in-altitude air density implied – please mention (same concerns the assumption that $w_{\text{sed}}$ is constant

**p7/l24** the two Smolarkiewicz papers list several flavours of MPDATA, which is used?

**p7/l26** "some value" $\rightsquigarrow$ "the value"?

**p7/l30** why 0.5?

**p8/l10** "Unlike to" $\rightsquigarrow$ "Unlike in"

**p9/Fig 2** caption: "Wellmixed" $\rightsquigarrow$ "well-mixed" (twice)

**p10/l4** is there any added-value in including the time loop in the pseudocode (same concerns listing on p13)

**p10/l5** indicate that only for quadratic sampling

**p10/Alg. 1 caption** random number interval: [...]

**p11/l11** random number interval: [...]

**p12/l29-31** use a proper big-oh (e.g., $\mathcal{O}(n)$ with `$\mathcal{O}(n)$`)

**p13/Alg. 2 caption** random number interval: [...]

**p14/l2** use a proper big-oh; nz inside the oh?

**p14/l18-21** needed?

**p15/l19** comment that constant air density implied

**p16/l12** remove "in the column model source code"

**p16/l16** omit "Validation exercises" from the section title (entirety of the paper is a validation exercise)

**p17/Fig. 4** "at the indicated points in time" – cannot see any indicated points in time

**p18/Fig. 5** ditto

**p18/Fig. 5** "use different y-axis" – cannot see different y-axis

**p18/l2** first mention of "lucky droplets" calls for a reference

**p19/Fig 6** BoxModelEmul not mentioned before, used only in figure captions

**p19/l6** "surprisingly well" – please be more specific

**p19/l7-8** "usually more easily reached" – please be more specific

**p19/9-10** "Even though..." – suggests this material can be skipped

**p19/l12-14** sounds like a sentence for Introduction or Conclusions

**p19/Fig. 7** no units for $dV$ (BTW, shouldn't it be $\Delta V$?)

**p20/l3** "according" $\rightsquigarrow$ "relevant"?

**p20/l8** avoid "We believe"

**p16/l31** "agreement ... is good" – please be specific, provide quantitative measure

**p21/l24** "In a technical experiment" – suggest rephrasing

**p21/l30-32** "Nevertheless..." – suggest skipping the sentence

**p22/l24** first mention of Long kernel without reference or comment

**p22/l30** rephrase around "now f or"

**p23/Fig. 8** no units for "dt" (BTW, shouldn't be $\Delta t$?)

**p23/l3** "occur too often" – please be more specific

**p24/l3-4** "has to be solved" – subcycling seems to me as a preferable option than changing timestep of the whole simulation ... suggest skipping/rephrasing the sentence

**p24/l4-5** please elaborate how/why inclusion of more SIP attributes would change the influence of LinSamp vs. QuadSamp choice?

**p25/Fig. 10** add ", respectively" at the end of caption

**p25/l13** "Wang" $\rightsquigarrow$ "Wang et al."?

**p25/l8** good place to mention performance trade-offs of the WM2D

**p25/l21** Bott/Bott's algorithm/model – please be consistent

**p27/l5** "Bott's results are reliable" – rephrase

**p28/l7&8** suggest renaming the section to "Algorithm profiling"

**p29/Tab. 1** use the same exponential notation as elsewhere (i.e., $A \cdot 10^B$ instead of $AeB$)

**p30/l6** "find their" – rephrase

**p30/l23** "For small SIPs j" – what is a small SIP?

**p31/l24** unit of "influx" should include 1/time

**p32/Fig. 15** avoid using two acronyms for the same thing: LS, LinSamp

**p33/Fig. 16** harmonise case for acronyms: Bin, BIN

**p33/l2** "collector SIP" – please elaborate

**p34/l4** "sounds like a banal ..." sounds too colloquial

**p34/l14** what is the former effect?

**p35/l4** "superiority ... in BIN" – rephrase

**p35/l12** "improvement of this" – rephrase

**p37/l14** same remark regarding riming, etc as for p3/l12

**p37/l11-12** please clarify if this statement concerns just this section

**p39/l11** "To bridge the gap" – puzzling, the gap was not mentioned earlier

**References**

Please harmonise the reference format:

- avoid double URLs (DOI URLs are unambiguous and enough): p41/l3-4, p41/l6, p41/l12-13, p41/l19-20, p41/l24-25, p41/l27, p41/l30-31, p42/l2, p42/l7-8, p42/l12-13, p42/l26, p43/l4-5, p43/l26-27

- add DOIs:

    - Kessler 1969: 10.1007/978-1-935704-36-2_1
    - Khairoutdinov and Kogan: 10.1175/1520-0493(2000)128%3C0229:ANCPPI%3E2.0.CO;2
    - Matsumoto and Nishimura 1998: 10.1145/272991.272995
    - Naumann and Seifert 2015: 10.1002/2015MS000456
    - Seifert and Beheng 2001: 10.1016/S0169-8095(01)00126-0
    - Shima et al. 2009: 10.1002/qj.441
    - Simmel et al. 2002: 10.1016/S0169-8095(01)00131-4
    - Smolarkiewicz 2006: 10.1002/fld.1071
    - Smolarkiewicz 1984: 10.1016/0021-9991(84)90121-9
    - Sölch and Kärcher 2010: 10.1002/qj.689
    - Wang et al. 2006: 10.1063/1.1928647
    - Wang et al. 2007: 10.1016/j.jcp.2007.03.029

- remove months: p41/l18, p41/l21, p41/l24, p41/l26, p41/l30, p42/l6, p42/l12, p42/l25, p43/l14, p43/l37, p44/l2

- remove ISSNs: p41/l24, p42/l26, p43/l14, p44/l2

Please correct:

- volume number in Grabowski et al. 2019

- page range for the work of Smoluchowski, should be 557-571 (see https://jbc.bj.uj.edu.pl/dlibra/doccontent?id=387533)

**Enclosed software**

I consider it awkward and confusing to use the GCC preprocessor with Python source code. Same concerns using csh and sed to generate preprocessor-directive-filled .py files with hardcoded system-dependent paths. Altogether, the multi-platform and work-out-of-the-box advantages of Python were lost.

Note that in the `CompSim.gcc.py` file you are using two independent random-number-generation infrastructures available in Python, and I doubt setting the seed via `random.seed()` affects values returned via `np.random.random()`.

Please avoid non-English comments in the code and marking changes with comments – it is the role of version control to track changes.

Please indicate in the code availability section:

- the code availability (or lack thereof), version and license for "Bott" and "Wang" models

- the license the LCM1D is released on

- the supported environments and dependencies of the implementation (python, numpy, gcc, csh, sed, ...)

Thank you for a useful contribution to the field!
Hope the above comments help,
Sylwester

**References**

[1] D.G. Dritschel et al. "The moist parcel-in-cell method for modelling moist convection". In: *Quarterly Journal of the Royal Meteorological Society* 144 (2018). DOI: 10.1002/qj.3319.

[2] S.G. Lasher-trapp, W.A. Cooper, and Alan M. Blyth. "Broadening of droplet size distributions from entrainment and mixing in a cumulus cloud". In: *Quarterly Journal of the Royal Meteorological Society* 131 (2005). DOI: 10.1256/qj.03.199.

[3] S. Shima et al. "Predicting the morphology of ice particles in deep convection using the super-droplet method: development and evaluation of SCALE-SDM 0.2.5-2.2.0/2.2.1". In: *Geosci. Model Devi. Discuss.* (2019). DOI: 10.5194/gmd-2019-294.

---

## Referee Comment (RC3) · Shin-ichiro Shima (Referee) · 22 Apr 2020

Review of "Collection/Aggregation in a Lagrangian cloud microphysical model: Insights from column model applications using LCM1D (v0.9)" by Simon Unterstrasser, Fabian Hoffmann, and Marion Lerch.

I would like to recommend this paper for publication after my concern is addressed, but which may require a major revision.

In this study, performance of several collection/aggregation algorithms for Lagrangian cloud models (LCMs) and bin models are compared in detail. The assessment was conducted in a one-dimensional columnar domain. Compared to their previous study in a single grid box, the tests are more relevant to three-dimensional cloud simulations, and hence the results are insightful.

My major concern is the limiter introduced in p.15, l.5. I think this may considerably diminish the performance of WM3D LinSamp. Instead, I suggest that the authors consider splitting the SIP, as detailed below.

**Major Comments**

1) p.15, l.5, limiter
   Now the limiter is implemented as follows: If $\nu_{coll} > \nu_j > \nu_i$,
   $$\nu_i' = \nu_i, \quad \mu_i' = (\nu_i \mu_i + 0.99 \nu_j \mu_j)/\nu_i,$$

   $$\nu_j' = 0.01 \nu_j, \quad \mu_j' = \mu_j.$$
   I think this is not a good idea because this could oversample small droplets. Instead, I recommend you to split the SIP $i$ as follows.
   $$\nu_i' = \nu_j' = \nu_i/2, \quad \mu_i' = \mu_j' = (\nu_i \mu_i + \nu_j \mu_j)/\nu_i.$$
   With this procedure, we can use more SIPs for large droplets. Then, we can expect that the number of limiter events $N_{LI}$ decreases because $\nu_{coll}$ tends to become smaller though the weighting factors of small SIPs ($\nu_j$) are not changed. Note that similar procedure is already incorporated in Shima et al. (2009) (see (5b) on p.1313), but this is not the same because weighting factor (multiplicity) is considered as integer in Shima et al. (2009) and therefore (5b) rarely happens.

**Minor Comments**

2) p.4, l.17; p.26, Fig.12; collision efficiency
   It is not clear which collision efficiency you are using for the default case.

3) p.14, ll.31-23
   The feature that each SIP does not appear in two pairs enables parallel computation. Somewhere in the paper, this favorable property of WM3D LS should be mentioned.

4) p.29, Table 2

From this result, I would conclude that WM3D LS is the most efficient.

5) p.31, Eq.(28c)

From this equation, we can derive the expected number of SIP pairs that actually collide:

$$N_{SIP}^{coll} = N_{combs}\overline{p_{crit}} \propto N_{SIP}^{\alpha+\beta-1}\delta t = N_{SIP}\delta t,$$

i.e., $N_{SIP}^{coll}$ is proportional to $N_{SIP}$. This would imply that linear sampling is reasonable.

**References**

Shima, S., Kusano, K., Kawano, A., Sugiyama, T. and Kawahara, S. (2009), The super-droplet method for the numerical simulation of clouds and precipitation: a particle-based and probabilistic microphysics model coupled with a non-hydrostatic model. Q.J.R. Meteorol. Soc., 135: 1307-1320. doi:10.1002/qj.441

---

## Author Comment (AC1) · 6 Aug 2020

**Reply to all three reviews**

The manuscript has been substantially revised. There are four main points:

1. The number of figures in the main body of the article is reduced from 23 to 15.
2. AON Linear Sampling produces better results with the recommended limiter condition.
3. The code has been polished (no csh, no sed).
4. A supplement with additional material is now included.

Authors of this paper test the behaviour of a numerical algorithm representing collisions between cloud and rain drops in Lagrangian microphysics schemes. They describe 3 versions of the algorithm (one of the versions is new) and test the convergence of the 3 versions in 3 different settings. The presented work is based on a similar suite of tests carried out by the same group of authors and published in GMD in 2017 (https://www.geosci-modeldev.net/10/1521/2017/gmd-10-1521-2017.html). In 2017 the tests were done in a 0-dimensional box setup. Now the tests are extended to a 1-dimensional column. I think that the presented tests are useful and the topic is interesting to the GMD community. However, the overall presentation of the convergence of the algorithm should be improved before publication. I couldn't find what is the criterion for reaching convergence in the tests. Despite being very thorough in testing different parameters and cases, the authors then use vague terms like "equally bad", "tend to approach the reference" or "seem to converge". The only way in which the authors show convergence is by plotting many lines on top of each other. This is not satisfying and results in a paper that is overflowing with figures that look the same. I suggest introducing a definition and quantitative measure of convergence. This would allow to change some of the repetitive concentration plots into plots showing convergence as a function of a tested parameter. I think that the paper should be more concise. The authors carried out a lot of tests but summarising some of them would improve the paper. For example Fig 10 shows results for slightly different initial condition and Fig 12 for a different kernel. Neither of these figures show anything new about the collision algorithm behaviour.

We agree that there are many similarly looking plots. With that we wanted to keep it simple and help the reader to interpret the results more easily when the same plot format is used. Moreover, this eliminated the need of introducing metrics which are sometimes more difficult to interpret.

On the other hand, we see that the manuscript is quite long. Hence, we follow your recommendation and identified sensitivity series where it is sufficient to show, e.g., only the moments after 60 minutes. The full time series plots are then moved to the newly introduced supplement. Moreover, the time evolution is shown for the range [30min,60min], where appropriate.

Similarly, the authors study the bin collision algorithm with upstream advection scheme and observe that the results are "slightly smeared out". The upstream advection scheme is known to be very diffusive and there were many papers published on that. This detracts from the main theme of the paper and the interesting parts of the study.

In previous publications, I faced sometimes concerns about EULAG/MPDATA as it based on the "infamous" upstream scheme. Hence, including both US1 and MPDATA should demonstrate that MPDATA does a better job. Nevertheless, we decided to remove all content that is related to the usage of the first order upstream scheme. The comparison with LCM should suffice to show the adequacy of MPDATA. By the way, we moved the PureSedi-test case in the previous section 3.1 to the Appendix.

Additional comments:

Table 1: German language in the caption

Sorry. Removed.

Page 3, line 13-14: "Moreover, we will use the term cloud droplets interchangeably with ice crystals to increase clarity in writing." - I don't think that this increases the clarity. The paper nowhere actually discusses issues related to ice crystals. I think that keeping the language focused on cloud and rain droplets is sufficient.

We removed the sentence and use the term droplet throughout the study. On the other hand, the text makes now more references to ice aggregation.

Figure 1: in SIP (Simulation particle) p should also be capitalised

Done.

Page 7 line 15: Smoluchowski equation

Thanks.

Page 8: Maybe it would make more sense to first describe the collision algorithms and then talk about the column model setup?

We would like to keep it our way. We think that first the basic properties of the column model should be introduced before going into the details of how collisional growth is implemented.

Figure 2,3 and later in the text: The abbreviations like AON, WM3D, etc were already defined and should not be defined again.

A table with all model versions and abbreviations is now included.

Page 9 line 5: I don't think it's legally allowed and generally acceptable to copy verbatim paragraphs from different papers (?). I would suggest to just refer to the relevant paragraph or to paraphrase.

We found it awkward to paraphrase our "own" text and thus we decided to copy paragraphs from our previous paper (it is also a GMD study). We were unsure if this plagiarism and we mentioned this when we submitted the manuscript. During the revision, we asked again the Editorial office and they confirmed that it is okay.

Algorithm 1 and 2 caption: The style convention of the code block should be repeated in its caption.

We moved the paragraph with the style convention to both captions.

Algorithm 2, line 16: iff

"iff" is a common term in mathematics meaning "if and only if". "iff" is replaced to avoid confusion.

Page 14 line 15: "For more sophisticated kernels, including, e.g., turbulence enhancement, the present approach may not be adopted easily as the driving mechanism for collisions to occur in the current model is differential sedimentation (...)." - This is very important for real applications of WM2D algorithm. Could you expand on this? How would you implement the WM2D ideas in a full 3D LES simulation focusing on turbulence effects on precipitation formation?

The extent of aggregation in pure ice clouds is not as much affected by turbulence as coalescence in warm clouds. So for aggregation studies with a classical hydrodynamic kernel, WM2D will be a suitable choice. The present WM2D version considers only vertical overtakes and can only be used with kernels that have a |wi-wj|-term. "Turbulence enhancement" kernels also account for collisions in the horizontal plane and such kernels usually do not have an

explicit |wi-wj|-term. So there is no straightforward way to use WM2D with such kernels and it is also not meaningful if a substantial fraction of collision appears due to turbulent motion. Moreover, this would also require making assumptions about the subgrid turbulent velocity of each SIP. That's another thread of research, namely, how subgrid information/processes are or should be incorporated in LCMs.

Paragraph 2.5: Please provide a table which summarises the combinations of different algorithm options and the labels used to distinguish them.

Included.

Paragraph 2.5: What is the added benefit of comparing the BIN Bott and Wang collection algorithms in a paper about Lagrangian collection algorithms?

The benefit is to put the comparison into a broader context. Bott and Wang are established solvers and it is important to see how differences between those two BIN models relate to differences among LCM versions or differences between LCM and BIN.

It also shows that Wang needs much smaller time steps than LCM for example. See also our reply on Figure 7 below.

Page 19 line 8: "We found that convergence is usually more easily reached for higher moments than for lambda_0" – Why is that? Would different method of initialising the SIPs make a difference?

These are good questions. And there is no simple answer. A few thoughts:

- In the base case, the decrease in lambda_0 is 4 orders of magnitudes, whereas lambda_2 increases by 8 orders of magnitude. This makes it difficult to define an objective metric that compares convergence in the zeroth and second moment. Hence, the above statement of an easier convergence is a bit subjective. So a possible answer to your question "Why is that" shouldn't be overrated.
- Our 'subjective' statement is mainly based on our experience gained from the 2017-paper. In particular, the supplement of the 2017-paper displays time evolutions of lambda_0, lambda_2 and lambda_3. There are several cases, where lambda_2-evolutions match better than lambda_0-evolutions. One example is Fig. 14 in SUPP of the present paper. There one can also find out about the effect of the various SIP initialization techniques.

Figure 7 and others: Could you show results for t> 30min only? The first half of all those plots shows nothing

The legends in the left column need some space, but the panels in the right column are shortened now. Thanks.

Figure 7 and others: Why are the Lagrangian schemes always predicting higher concentrations than the BIN scheme?

Good question. In U2017, we saw that the converged AON results predicted slightly smaller concentrations after 60 minutes. Dziekan & Pawlowska, 2017 (later in the year 2017) pointed out that AON results, at least in the limit of all SIP weighting factors approaching unity, solve the so-called master equation, and not the mean-state equation of Smoluchowski. This gave a posteriori one possible physical explanation of why AON and BIN results can differ and why there are more collisions in AON. In the end, the story is a different one, as we had a small bug in the 2017-program code. The present study uses a bug-free algorithm and AON produces fewer collisions. By the way, this gave the motivation for Figs. 9-11 (in the original manuscript),

where we wanted to see if this is a universal feature of AON and if this occurs also for other LWC- and DNC-values. Moreover, we compare LCM results to one specific BIN model. It could be that the underlying Bott model is diffusive in radius space and overestimates the amount of collisions. This is why we alternatively used the algorithm by Wang and find it important to include the fourth row of Fig.6 with bin sensitivities.

Figure 14: This figure is very interesting! It's staggering to see how very few of the tested combinations of SIPs lead to any collisions. A strong argument in favour of WM2D approach.

Indeed, very few of the tested combinations of SIPs lead to any collisions.

Figure 16: The figure is not clear, especially the yellow dash-dotted lines are hard to see.

We changed the selection of colours.

Page 34 lines 15-19: If the results are "identical" and "basically identical" then they don't have to be shown again.

We moved it to SUPP.

Page 35 line 4: This is not a new finding and is out of scope of the paper.

As written above, we removed all first-order upstream results in the collisional growth sections except for Figs. 4 & 5 (now Figs. A1 & A2).

Figure 21: It's very hard to distinguish between the two blue lines.

We now use a different colour table.

Page 40 line 14: necessary is repeated

Thanks.

Code availability: It would help to provide a Docker or a Singularity image in which the column model scripts could be run. It would eliminate the need to change any file paths or to install compilers and packages with specific versions. This would be helpful, especially because the provided code is not just pure Python code. I know that it is not a policy of GMD, so it's just a suggestion. In my case I tried compiling the code with gcc 8.3.0 and gcc/9.2.0 on a CentOS system with Python 3.7.0 and got an error:

AON_Alg.fpp:90:0: error: operator '*' has no right operand #if (KERNEL_INTPOL <= 1) /* logarithmic mass bin*/

I suspect it is some preprocessor issue but I didn't debug further. I then tried compiling with gcc/9.2.0 on an OSX system with Python 3.7.6 and got an error: sed: illegal option –

I guess it's an issue with OSX default sed, but I didn't debug further

The present study is my first open-source endeavour, and in the meantime I have come to the conclusion that not all aspects of the code design helped the interoperability. To give more tribute to the FAIR principles, I replaced the csh-script by a python script. Calls of sed are replaced by python internal commands (module re). So no more csh, no more sed. Unfortunately, gcc could not be replaced by python constructs. There are some modules promising to do the job of gcc, but they do not seem to be maintained or got stuck during their development.
* * *
Hereby I provide my comments to the manuscript entitled \Collection/Aggregation in a Lagrangian cloud microphysical model: Insights from column model applications using LCM12 (v0.9)"

Overall, I consider the submitted text a valuable contribution to the literature on particle-based cloud modelling as it provides detailed formulation of test cases that are essential in the development of new implementations of relevant algorithms. It thus clearly matches the journal scope.

Below, I list my major, minor and technical comments to the manuscript. I include also a comment to the enclosed software.

**Major points**

**Unacknowledged performance trade-offs**

Some performance trade-offs pertaining to the choice among linear and quadratic sampling are detailed. It is however not pointed out that QUADRATIC sampling precludes parallelisation of the collision computations (within a gridbox/column) due to introduced data dependency (page 11, line 22-23). This is particularly worth underlining, as the availability of shared-memory parallelisation with multi-core CPUs or GPUs allows for significant speed up (i.e., almost by the factor equal to the number of threads). In fact, all but the pair-shuffling and random number generation steps in the AON coalescence algorithm with linear sampling are embarrassingly parallel. Given the above, I find it at least misleading to say, without mentioning the precluded parallelisability, that:

• \simulations with linear sampling ... converges slower ... compared to quadratic" (p1/l14-15)

• \benefit of the reduced computational cost may be outweighed by the stronger requirements on $\Delta t$" (p24/l2)

• \restrictions on the timestep might cancel out the computational benefit gained by the reduced number of SIP combinations" (p39/l25-26)

Similarly, I doubt the statement on page 12, lines 2-3 (on performance superiority over integer preserving implementation) holds true in parallel context, where random numbers can be generated concurrently in large batches.

Finally, it is worth commenting on the parallelisability consequences of the requirement to perform collisions column-wise in the WM2D scheme.

The new LinSamp implementation with a new limiter version performs much better and statements that you found too critical are removed. Furthermore, comments on parallelisation are included in the LinSamp and WM2D sections.

**Subrid-dynamics and WM2D**

While it is acknowledged on page 14 (lines 15-17) that the WM2D scheme is somewhat incompatible with \sophisticated kernels", I highly recommend to extend the discussion also to the aspects of subgrid-scale dynamics representation in particle-based models { e.g., referring to the already cited work of Hoffmann et al. (2019). In short, in my understanding, the \information content" of SIP positions in particle-in-cell-type models is somewhat overestimated here. In particular, the prevalent Large-Eddy-Simulation context should be addressed. Candidate location: page 14 (lines 11-14)?

It is clear, that the complexity of AON when employed in 2D/3D cloud LES is in between of highly parametrised bulk approaches and DNS computing trajectories of single droplets. AON with WM2D is an effort to use slightly more information of the SIPs (i.e. their vertical position) than in the regular AON.

A possible route to consider the effects of subgrid-motions on collision in LCMs has recently been presented by Krueger and Kerstein (2018, https://doi.org/10.1029/2017MS001240). Their one-dimensional approach is able to represent droplet clustering and turbulence-induced relative droplet velocities in a realistic manner, and its implementation in already applied LCM subgrid-scale models (e.g., Hoffmann et al. 2019) is deemed straightforward. However, further research is required on how the limited number of SIPs in current LCM applications may corrupt the correct representation of such processes. We include a paragraph in the manuscript.

Rebutting your argument "positions in particle-in-cell-type models is somewhat overestimated here": If (subgrid) position of SIPs should not be overinterpreted, then it would not make sense to interpolate background Eulerian fields to SIP positions as is done in Grabowski et al 2018, GMD or GMDD paper of Shima et al., 2020.

**Paper length**

The article length is, in my opinion, impeding appreciation of its content. I include some detailed suggestions on what could be omitted from the text in Technical/editorial remarks below. Besides that, I consider it a malpractice to introduce an almost-page-long quote from an earlier study of the authors. I see also little benefit in repeating Figure. 3 here { please just refer to the relevant parts of the 2017 GMD paper which is readily available for all readers. I also suggest adding a table of contents (as done recently in GMD in Shima et al. [3]).

If I had cited the text of someone else in this way, I would consider it malpractice. In the present situation, it is fairer to simply copy the text. There's no benefit of rewriting it. As replied to Rev #1, the Editorial Office confirmed that it is legally okay.

The left side of Fig.3 was not part of the previous 2017-paper. Moreover, the two other algorithms (which performed mediocrely in the 2017-paper) are not actively used any longer (to our knowledge). So it makes sense to use this updated sketch in the future.

GMD Editorial Office is reluctant when it comes to including a printed table of contents. They say the PDF document contains a table of content. We drastically shortened the number of figures and hope that readers do not get lost that easily anymore.

**Minor points**

**The title**

First, why not to avoid a \slash" in the title, and use \Collisional growth" instead of \Collection/Aggregation". Second, I oppose to calling presented simulations \applications", suggest \simulations"?

Finally, I generally suggest to label the discussed microphysics modelling methods as \probabilistic particle-based" rather than \Lagrangian". First, \Lagrangian" is a much more broader term (consider e.g. the Lagrangian cloud models described in [2, 1]), and thus potentially misleading for readers from outside our niche. Second, the discussed model is not fully Lagrangian as it relies on Eulerian dynamical core. I am aware that the present title is a reference to the U2017 paper, but perhaps the above arguments outweigh it? On a related note, there is not a single mention of the \Monte-Carlo" keyword in the paper, please do cater to a wider community and use such keywords to give a good context for the readers.

Indeed, we wanted to stress the analogy with the 2017-paper. But we agree with your perspective. We will switch to the terms "collisional growth" and "particle based" in the title.

AON can be interpreted as a Monte-Carlo method as we average over 20 independent simulation realisations. Once AON is employed in higher-dimensional models, it is not foreseen to run multiple realisations in each grid box. Then, averaging is supposed to occur across grid boxes. In this context I would not call it a Monte-Carlo method. Then probabilistic method is a more appropriate term.

**The limiter**

The ad-hoc definition of the \limiter" (p15/l6) calls at least for a reference to the min() in the SDM paper's ~ $\gamma\alpha$ := min($\gamma\alpha$; [$\xi j\alpha$; $\xi k\alpha$]) expression (Shima et al. 2009, step (5) in the left column), if not for reformulating the \limiter" in a more robust manner.

Rev# 3 (S. Shima) had a similar comment and proposed to try his limiter variant. We carried out thorough tests with different limiter implementations. It was interesting to see how a seemingly small detail of an algorithm can have a large impact on the performance. We go into more detail in the reply to Rev #3.

**Integer vs. real-valued weighting factors**

I do not find enough grounds in the text for the statements on the superiority of real-valued vs. integer-valued weighting factors. Besides the above commented issue of parallel random number generation/multiple collisions, the statements on page 11, lines 30-31 seem to overlook the concept of spectrum estimation, see e.g. the third paragraph in section 5.1.4 in (Shima et al., 2009) (also, worth mentioning when discussing eq. 14).

Dealing with integers and preserving integer values for the weights needs some additional steps in the algorithm formulation. The number of FLOPS for computing a random number is high relative to the amount of computations in the remaining parts of the AON algorithm. In my opinion, dealing with integers makes sense in small discrete examples (as in the cited reference Dziekan & Pawlowska, 2017). In a setting, where SIP weights exceed, say 1e6, it is more appropriate to use floats. For me, nu_i= 3523496 pretends exactness which is not given.

Independent of these considerations, the question remains how one wants to reconstruct or derive physical quantities from a discrete SIP ensemble. Shima et al propose a statistical method called spectrum estimation where a continuous DSD is constructed from a discrete SIP ensemble. But I do not see how this is related to the aspect of integer/float weights.

**CFL condition for sedimentation**

I would argue that we should assume the CFL condition for particle sedimentation as well -- while it does not cause the numerics to fail as in Eulerian component, it is intuitively not fulfilling the assumptions (as acknowledged on p4/l12). Relevant statements: p8/l11, p12/l26, p21/l15, p34/l10.

In our current LCM model system only sedimentation and collisional growth are considered and it is perfectly fine to choose time steps that are larger than CFL. But we agree that in higher-dimensional models that include diffusional growth choosing a CFL-limited time step seems more appropriate. This holds in particular for warm clouds. From my experience, time step choice in a typical cirrus simulation is not that crucial. This depends on the time scale of deposition and also on the smoothness of the background fields.

At least, it is good to see that in the present model system fairly large time steps produce reasonable results. So time step requirements of collisional growth alone do not seem to be a bottleneck in LCMs. Including diffusional growth and its interaction with collisional growth process may, however, demand collisional growth time steps as small as those for diffusional

growth.
A discussion on this is included in the revised version.

**Courant number values in section 3.1**

Upwind and MPDATA convergence is dependent on the Courant number -- please indicate which was used in section 3.1. Perhaps worth checking the behaviour for a set of Courant numbers.

As stated in section 2.2 the maximum CFL number is 0.5. This was also used in section 3.1 (it is now part of the Appendix). Each bin has its own "local" CFL number that depends on $w_{sed}$, $\Delta z$, $\Delta t$ and $r_{CFL}$. The figure included in this reply shows the bin-dependent local CFL number (top) and the number of subcycles (bottom) to maintain CFL numbers below $r_{CFL}$. It becomes obvious that for most bins the local CFL number is not close to $r_{CFL}$. Hence prescribing a theoretically advantageous value of $r_{CFL} = 0.5$ has no real practical consequences.

The supplement now discusses a variation of $r_{CFL}$ and $\Delta t$ (both parameters take values as in the plot in this reply) for BIN solutions for HalfDomLinDec setup. We find that the BIN solutions do not depend on the tested parameters.

**Correlations**

Please be more specific as to the mathematical meaning of \correlations" mentioned 12 times in the context of collisions but never defined.
We use the term correlation always in the same context, so there is no ambiguity in this respect. At several occasions we refer to the relevant literature (papers by first authors Bayewitz, Gillespie, Alfonso and Wang for example). Readers not familiar with the master equation, statistical fluctuations and correlations in the context of collisional growth are strongly advised to read the aforementioned publications to gain a deeper understanding of the matter. We believe this cannot be achieved in the present paper.

**Classical/regular nomenclature**
References to \classical/regular" \implementations/approaches/cases/versions/AON" (p8/l15, p8/l16, p8/l22, p8/l24, p9/l4, p11/l12, p12/l8, p12/l19, p12/l29, p12/l31, p14/l7, p14/l8, p14/l30, p15/l4, p17/l2, p19/l1, p25/l5, p25/l21, p29/l1, p30/l2, p37/l12) are not understandable, especially given that the authors introduce their own nomenclature for numerous notions named differently in literature. I am aware that there are some definitions of \regular" in the text, but it is an over 40-page long paper. Please come up with more precise and less subjective statements.

A table listing all model versions is introduced in order to have a clear location where everything is defined. The terms "implementations/approaches/cases/versions/variants" were used more or less interchangeably but with slightly different flavours. Apparently this caused more confusion than it helped. The revised manuscript contains a cleaner terminology.

Anyhow, regular AON always referred to the AON version described in section 2.3.1. The title of the section is "Regular AON collection algorithm (WM3D)". So I do not see that this is subjective in any way.

**Technical/editorial remarks**
**Figures**

I urge the authors to replace raster images in figures 1 and 4-23 by its vector-format equivalents (i.e., plt.savefig(format='png', ...) ; plt.savefig(format='pdf', ...)). I suggest using *cm*-3 as the unit for $\lambda 0$ on the plots.

We switch to pdf versions of the mentioned figures. I favour units m$^{-3}$.

**Text**

**p1/l2** the \high number" is equally (or even more) applicable to bin models, please rephrase and indicate with respect to which benchmark the value is high.

Clarified.

**p1/l4** is the word \explicit" needed (suggest avoiding if the opposite \implicit" is not clear)

Removed

**p1/l8** ditto

Removed

**p1/l12** \accuracy" ; \resolution"

Changed it to resolution.

**p2/l28** please underline that it is you who introduce the AON term

Done.

**p2/l22** why not in chronological order? (see also background works listed in Shima et al. [3] and dating back to 2004)

No real reason, why it is not chronological. Corrected.

If you refer to Paoli et al, 2004, it is true that they used a particle-based approach for modelling contrails. I did not add the paper to the list, as their model does not include processes of natural cirrus formation and hence its applicability is limited to the one specific purpose of the paper.

**p2/l26** \abbreviated as ... in the following" ; \hereinafter abbreviated ..."?

Done.

**p3/Table 1** please remove non-English caption, and consider removing the table { given the multitude of symbols used in the text, it seems anecdotal to list 8 abbreviations in a table (moreover, the following are not listed: DNC, MPDATA, CFL, WM2D, WM3D, MC, BIN, US1, noSedi, LCM0D, LCM1D, LS, ...)

Table 1 is updated and lists frequently used abbreviations. Abbreviations with a local scope are not added.

**p3/l3** suggest removing/rephrasing \relatively young modelling approach" { particle-in-cell method is 50-year old; same for Monte-Carlo for collision

added "in cloud physics" to make the statement more precise.

**p3/l4** \aws" ; \was"

Done.

**p3/l12** \coalescence, aggregation, or accretion" { mention \riming", \self-collection", \wash-out" keywords as well?

These terms cover the general interactions of liquid-liquid, ice-ice, and ice-liquid hydrometeors, which include the interactions listed by the reviewer. To be slightly more inclusive, we changed our text to: „We will use the term collision, including various processes such as coalescence, aggregation, or accretion, as we focus on […]"

**p3/l13-14** actually, ice crystals are mentioned only one after this statement { suggest removing

The text makes now more references to ice aggregation.

**p3/l22** \(sometimes pedantically)" sounds negative ... suggest not being pedantic (see below) and removing the statement

Done.

**p4/l3** suggest using subscripts for *Lz* and *nz*
Done.

**p4/l6** no need to make it a separate numbered equation?
The reason for prominent placement is that $\Delta A$ is important for WM2D and a change in $\Delta z$ implicitly changes $\Delta A$ if $\Delta V$ is kept constant.

**p4/l8** no need to define volume of the sphere - just mention in the text

That's why I have written pedantic in the beginning☺. Even though it seems clear, the point is that in ice microphysics, this expression is usually replaced by mass-size relationships of the type $m = a\, L^b$. So in follow-up papers considering more specifically aggregation, it helps to say that Eq. (3) of the original formulation is replaced.

**p4/l12** *K*(*mi; mj*) and *K*(*ri; rj*) would likely be better named differently
True.

**p4/l15** \radius-dependent" ; \sie-dependent"
Done.

**p4/l17** \latter" ; \last" (there are three assumptions listed before)
Corrected.

**p4/l18** there is a bogus character before 500 $\mu$m - garbage displayed in the pdf viewer I'm using
I cannot reproduce your error. The symbol is "\gtrsim" (in latex slang)

**p5/l1** \collection" ; \collisional growth"?
Changed.

**p5/l9** \latter" ; \last" (there are three assumptions listed before)
Corrected

**p5/l12** no need to define factorial
Okay.

**p5/l13** skip reference to Berry 1967, surely \mass density function with respect to the logarithm of radius" is enough
Those who are not familiar with the definition will be happy about the reference as it is nicely explained in Berry, 1967.

**p5/l22** first sentence is a repetition from the Introduction
From this sentence on, section 2.1 treats LCM-specific definitions and settings. So one sentence that introduces LCM (for those who are not familiar with particle-based approaches) should be okay.

**p5/l24-25** suggest rephrasing around \terms low and high"
Hopefully clearer now.

**p6/l14** random interval should be [...) and not [...] (as per numpy.random docs)
Changed it.

**p6/l14** \some threshold" { please be specific (also, worth mentioning the alternative formulation with integers)
"singleSIP-init" in U2017 describes it in more detail and also specifies the threshold.

**p7/l3** \around" { please be specific
I do not really see your point. Eq. 16 gives a good rule of thumb. By the way, panel c) of Fig.5 lists both N_SIP and kappa. You can convince yourself that the factor 5 is a good approximation. I do not think it helps to be more specific as this factor depends on DSD properties. Or should I list the precise scaling factor for each kappa-value? But please be aware, that the initial N_SIP

can change by 1 or 2 across the different realisations. The listed kappa-values in the figure are actually rounded.

**p7/l6** random number interval: [...]
Changed it.

**p7/l6-7** unneeded sentence (Furthermore...)?
Why not needed? It is never stated explicitly elsewhere.

**p7/l15** rephrase \to solve the Smoluchowski."
Done.

**p7/l18** use partial derivatives
Thanks. Corrected.

**p7/l21** constant-in-altitude air density implied: please mention (same concerns the assumption that $w$sed is constant
The only place, where rho$_{air}$ matters, is in the computation of the fall speed using Beard's formula. We now state that rho_air = 1.225 kg/m³ is used, consistent with Bott's and Wang's specification.

The column model computations, in particular solving the sedimentation/advection part, do not need the information on air density. In the current setup, the subtlety between advecting mixing ratios and concentrations is not relevant here for hydrometeors (unlike to gases, which we do not treat here). But we now state clearly in the text, that we assume constant-in-altitude air density.

**p7/l24** the two Smolarkiewicz papers list several flavours of MPDATA, which is used?
We use the basic MPDATA. For the given advection problem (non-divergent, constant in time advection speed), no extensions like infinite-gauge, variable-sign, non-oscillatory options had to be used. We added this piece of information to the manuscript.

**p7/l26** \some value" ; \the value"?
Done.

**p7/l30** why 0.5?
MPDATA has optimal performance for CFL number of 0.5. However, from the plot included here it should become obvious that the practical consequences of choosing 0.5 as upper threshold r$_{CFL}$ are not too large. Having in mind that the bin-dependent CFL numbers can be far from the prescribed rCFL-value, we refrain from stating in the manuscript that r$_{CFL}$=0.5 is the optimal choice. This would give a wrong impression.

Description of the Figure:

The upper panel shows the bin-dependent CFL number for six different simulations. The y-axis uses a mixed linear/log-scale. The solid/dotted lines show simulations with Δt = 2s and 10s, respectively. The three colours denote r$_{CFL}$ (in the inserted legend the quantity is called CFLmax). In the left part of the bin grid the local CFL number are much smaller than

$r_{CFL}$. In the large-radius region the curves exhibit a saw-tooth pattern. Each time the number of subcycles is incremented (see bottom panel), the local CFL number drops.

SUPP now includes a sensitivity study where BIN simulations of the HalfDomLinDec setup were repeated with the $\Delta t$- and $r_{CFL}$ -values used. It had no effect on the model outcome.

**p8/l10** \Unlike to" ; \Unlike in"
Done.

**p9/Fig 2** caption: \Wellmixed" ; \well-mixed" (twice)
Done.

**p10/l4** is there any added-value in including the time loop in the pseudocode (same concerns listing on p13)
Is there any added value of removing it? I prefer to make clear that AON uses a prescribed time step. One cloudl also think of designing algorithm where the time advancement is computed based on the collision probability.

**p10/l5** indicate that only for quadratic sampling
Nowhere is claimed that the presented pseudo-code treats the linear sampling version. So I do think adding 'quadratic sampling' confuses more than it helps.

**p10/Alg. 1 caption** random number interval: [...]
Done.

**p11/l11** random number interval: [...]
Done.

**p12/l29-31** use a proper big-oh (e.g., *O*(*n*) with $\mathcal{O}(n)$)
Thanks.

**p13/Alg. 2 caption** random number interval: [...]
Done.

**p14/l2** use a proper big-oh; nz inside the oh?
Done.

**p14/l18-21** needed?
I prefer to keep it in order to see nicely the differences between both approaches.

**p15/l19** comment that constant air density implied
See comment above (p7/l21)

**p16/l12** remove \in the column model source code"
I wanted to make clear, that the new box model results (noSedi) also rely on the same source code.

**p16/l16** omit \Validation exercises" from the section title (entirety of the paper is a validation exercise)
Done.

**p17/Fig. 4** \at the indicated points in time" { cannot see any indicated points in time
I am confused. Figs. 4 & 5 both use a legend telling you which points in time are displayed.
**p18/Fig. 5** ditto

I am confused. Figs. 4 & 5 both use a legend telling you which points in time are displayed.

**p18/Fig. 5** \use different y-axis" { cannot see different y-axis
I am confused. The plots in Fig. 5 use different y-ranges!?

**p18/l2** first mention of \lucky droplets" calls for a reference
Done.

**p19/Fig 6** BoxModelEmul not mentioned before, used only in figure captions
Solved.

**p19/l6** \surprisingly well" { please be more specific
Changed to "Hence, AON works well even for large time steps, […]"

**p19/l7-8** \usually more easily reached" { please be more specific
Changes to "Generally, we find faster convergence for higher moments than for \lambda_0 (not shown)."

**p19/9-10** \Even though..." { suggests this material can be skipped
Why? I think this is important for clarification.

**p19/l12-14** sounds like a sentence for Introduction or Conclusions
Moved to the beginning of 3.1.

**p19/Fig. 7** no units for $dV$ (BTW, shouldn't it be $\Delta V$ ?)
Corrected. Yes ΔV.

**p20/l3** \according" ; \relevant"?
I guess it is p21. Changed.

**p20/l8** avoid \We believe"
Okay.

**p16/l31** \agreement ... is good" { please be specific, provide quantitative measure
As mentioned in the reply to reviewer 1 introducing quantitative measures is difficult. And the interpretation based on a seemingly objective measure is again subjective.

**p21/l24** \In a technical experiment" { suggest rephrasing
Changed.

**p21/l30-32** \Nevertheless..." { suggest skipping the sentence
I do not see why this sentence should be removed.

**p22/l24** first mention of Long kernel without reference or comment
We now mention the Long kernel in the beginning of section 3.1.

**p22/l30** rephrase around \now f or"
Corrected.

**p23/Fig. 8** no units for \dt" (BTW, shouldn't be $\Delta t$?)
It should be Δ$t$. For consistency with other plots, units are added to the legend in Fig.4 (I believe you meant Fig.4, not Fig.8.).

**p23/l3** \occur too often" - please be more specific
Rephrased paragraph.

**p24/l3-4** \has to be solved" { subcycling seems to me as a preferable option than changing timestep of the whole simulation ... suggest skipping/rephrasing the sentence
Obsolete as paragraph has been removed.

**p24/l4-5** please elaborate how/why inclusion of more SIP attributes would change the influence of LinSamp vs. QuadSamp choice?
Obsolete as paragraph has been removed.

**p25/Fig. 10** add \, respectively" at the end of caption
Fig. 10 is not part of the manuscript any longer.

**p25/l13** \Wang" ; \Wang et al."?

**p25/l8** good place to mention performance trade-offs of the WM2D

Done here and also in section 2.3.1.

**p25/l21** Bott/Bott's algorithm/model - please be consistent

Corrected.

**p27/l5** \Bott's results are reliable" - rephrase

Changed to "While Bott yields stable results for Δt ≤ 100 s, the results only converge for Δt ≤ 20 s."

**p28/l7&8** suggest renaming the section to \Algorithm profiling"

Changed. This sounds definitely better.

**p29/Tab. 1** use the same exponential notation as elsewhere (i.e., $A \cdot 10B$ instead of $AeB$)

The notation with "e" saves space. Otherwise the table width is too large.

**p30/l6** \find their" { rephrase

Done.

**p30/l23** \For small SIPs ¡" { what is a small SIP?

Small SIP is defined in section 2.1.

**p31/l24** unit of \influx" should include 1/time

No value is given for the influx. So your comment is not applicable.

**p32/Fig. 15** avoid using two acronyms for the same thing: LS, LinSamp

We reduced the occurrence of LS.

**p33/Fig. 16** harmonise case for acronyms: Bin, BIN

Done.

**p33/l2** \collector SIP" { please elaborate

Rephrased.

**p34/l4** \sounds like a banal ..." sounds too colloquial

Rephrased.

**p34/l14** what is the former effect?

Rephrased.

**p35/l4** \superiority ... in BIN" - rephrase

The whole paragraph was re-written.

**p35/l12** \improvement of this" -rephrase

Rephrased.

**p37/l14** same remark regarding riming, etc as for p3/l12 p3/l12

Please check our previous comment on this aspect.

**p37/l11-12** please clarify if this statement concerns just this section

Clarified.

**p39/l11** \To bridge the gap" { puzzling, the gap was not mentioned earlier

Rephrased

**References**

Please harmonise the reference format:

I polished my paper reference database and all your suggested actions were implemented.

**Enclosed software**

I consider it awkward and confusing to use the GCC preprocessor with Python source code. Same concerns using csh and sed to generate preprocessor-directive-filled .py files with hardcoded system dependent paths. Altogether, the multi-platform and work-out-of-the-box advantages of Python were lost.

The new version v1.0 does not require sed and csh. However, gcc was not replaced as mentioned in the reply to Rev #1.

Note that in the CompSim.gcc.py file you are using two independent random-number-generation infrastructures available in Python, and I doubt setting the seed via random.seed() affects values returned via np.random.random(). Please avoid non-English comments in the code and marking changes with comments { it is the role of version control to track changes.

non-English comments were removed.
The seeding for np.random.random() was introduced.

Please indicate in the code availability section:

• the code availability (or lack thereof), version and license for \Bott" and \Wang" models
I obtained both BIN codes from the respective researchers. They are not published under a licence.

• the license the LCM1D is released on
Now included.

• the supported environments and dependencies of the implementation (python, numpy, gcc, csh, sed, ...)

The GitHub repository contains a document that lists the program/module versions with which I ran the simulations. The document has been updated as csh and sed are not mandatory any longer.

Thank you for a useful contribution to the field!
Hope the above comments help,
Sylwester

Many thanks for the thorough review, which triggered many improvements.

In this study, performance of several collection/aggregation algorithms for Lagrangian cloud models (LCMs) and bin models are compared in detail. The assessment was conducted in a one-dimensional columnar domain. Compared to their previous study in a single grid box, the tests are more relevant to three-dimensional cloud simulations, and hence the results are insightful. My major concern is the limiter introduced in p.15, l.5. I think this may considerably diminish the performance of WM3D LinSamp. Instead, I suggest that the authors consider splitting the SIP, as detailed below.

**Major Comments**

Now the limiter is implemented as follows: If $\nu_{coll} > \nu_j > \nu_i$,
$$\nu_i' = \nu_i, \quad \mu_i' = (\nu_i\mu_i + 0.99\nu_j\mu_j)/\nu_i,$$

$$\nu_j' = 0.01\nu_j, \quad \mu_j' = \mu_j.$$

I think this is not a good idea because this could oversample small droplets. Instead, I recommend you to split the SIP $i$ as follows.
$$\nu_i' = \nu_j' = \nu_i/2, \quad \mu_i' = \mu_j' = (\nu_i\mu_i + \nu_j\mu_j)/\nu_i.$$

1) p.15, l.5, limiter

I think this is not a good idea because this could oversample small droplets. Instead, I recommend you to split the SIP as follows.

With this procedure, we can use more SIPs for large droplets. Then, we can expect that the number of limiter events decreases because tends to become smaller though the weighting factors of small SIPs ( ) are not changed. Note that similar procedure is already incorporated in Shima et al. (2009) (see (5b) on p.1313), but this is not the same because weighting factor (multiplicity) is considered as integer in Shima et al. (2009) and therefore (5b) rarely happens.

This rather small detail drastically improved the performance of the LinSamp version. So large parts of the sections concerned with LinSamp have been re-written.

Moreover, LinSamp simulations were added to many further test cases compared to the original script. A thorough discussion of the critical limiter choice is added to the Supplement. I am glad, that no results of the "flawed" LinSamp algorithm appear in the final paper. This avoids confusion in the community.

 **Minor Comments**

2) p.4, l.17; p.26, Fig.12; collision efficiency

It is not clear which collision efficiency you are using for the default case.

Yes. We missed to write that the BoxModelEmul simulations used the Long kernel by default. This information is now included in the beginning of section 3.1.

3) p.14, ll.31-23

The feature that each SIP does not appear in two pairs enables parallel computation. Somewhere in the paper, this favorable property of WM3D LS should be mentioned.

Good point. It is mentioned now in section 2.3.3.

4) p.29, Table 2

From this result, I would conclude that WM3D LS is the most efficient.

Yes. Linear Sampling is the most efficient, but the table alone does not make a statement about effectiveness. Moreover, $N_{comb}$ alone is not a perfect metric for computational cost, as in WM2D the cost per tested combination is smaller. In the revised version (with much better performance of LinSamp due to the modified limiter), it will be made clearer that LinSamp is the most efficient approach. See especially the new paragraph at the end of section 3.2.

5) p.31, Eq.(28c)

From this equation, we can derive the expected number of SIP pairs that actually collide:

$$N_{SIP}^{coll} = N_{combs}\overline{p_{crit}} \propto N_{SIP}^{\alpha+\beta-1}\delta t = N_{SIP}\delta t,$$

i.e., $N_{SIP}^{coll}$ is proportional to $N_{SIP}$. This would imply that linear sampling is reasonable.

Many thanks for the derivation of the formula, but I doubt it is correct. Once $p_{crit} > 1$ occurs and multiple collections are implemented, the above formula is not valid as values $p_{crit} > 1$ should be rounded down to 1.

Imagine a hypothetical case with 200 combinations, where 100 combinations have $p_{crit} = 0.5$ and the other 100 combinations have $p_{crit} = 1.5$. Then your formula implies $N^{coll}_{SIP} = N_{comb}$. But the correct value is 100 * 0.5 + 100 * 1 = 150.

Kind regards,
Simon Unterstraßer, on behalf of all authors